# Autism spectrum disorder-like behavior caused by reduced excitatory synaptic transmission in pyramidal neurons of mouse prefrontal cortex

Hiroaki Sacai[1], Kazuto Sakoori[1,2], Kohtarou Konno[3], Kenichiro Nagahama [1,2], Honoka Suzuki[1,2], Takaki Watanabe[1,2], Masahiko Watanabe[3], Naofumi Uesaka [1,2,4✉] & Masanobu Kano [1,2✉]

Autism spectrum disorder (ASD) is thought to result from deviation from normal development of neural circuits and synaptic function. Many genes with mutation in ASD patients have been identified. Here we report that two molecules associated with ASD susceptibility, contactin associated protein-like 2 (CNTNAP2) and Abelson helper integration site-1 (AHI1), are required for synaptic function and ASD-related behavior in mice. Knockdown of CNTNAP2 or AHI1 in layer 2/3 pyramidal neurons of the developing mouse prefrontal cortex (PFC) reduced excitatory synaptic transmission, impaired social interaction and induced mild vocalization abnormality. Although the causes of reduced excitatory transmission were different, pharmacological enhancement of AMPA receptor function effectively restored impaired social behavior in both CNTNAP2- and AHI1-knockdown mice. We conclude that reduced excitatory synaptic transmission in layer 2/3 pyramidal neurons of the PFC leads to impaired social interaction and mild vocalization abnormality in mice.

[1] Department of Neurophysiology, Graduate School of Medicine, The University of Tokyo, Tokyo 113-0033, Japan. [2] International Research Center for Neurointelligence (WPI-IRCN), The University of Tokyo Institutes for Advanced Study (UTIAS), The University of Tokyo, Tokyo 113-0033, Japan. [3] Department of Anatomy, Hokkaido University Graduate School of Medicine, Sapporo 060-8638, Japan. [4] Graduate School of Medical and Dental Sciences, Tokyo Medical and Dental University, Tokyo 113-8510, Japan. ✉email: uesaka@m.u-tokyo.ac.jp; mkano-tky@m.u-tokyo.ac.jp

Autism spectrum disorder (ASD) includes a wide range of neurodevelopmental deficits and diagnosed based on impaired social interaction or communication, and stereotyped or repetitive behaviors. Hundreds of genes associated with ASD have been identified[1,2]. Many of these genes encode synaptic proteins at both pre- and post-synaptic sites including neuroligins, neurexins, cadherins, PROSAP/SHANK, synapsin, and neurotransmitter receptors[3–9]. Several animal models with mutations of these genes show synaptic dysfunctions and ASD-like behaviors in mice[10–15]. Thus, it has been thought that synaptic dysfunction underlies the pathophysiology of ASD[12,14–17]. Several mouse models of ASD are shown to display abnormal excitatory synaptic transmission and social deficits[12,13,18]. Modulation of AMPA-type glutamate receptor, the major excitatory neurotransmitter receptor in central neurons, is reported to rescue deficits in some ASD mouse models[19,20]. However, roles of many ASD-related genes in synaptic function and ASD-related behavior have not been investigated, and causal relationship between synaptic dysfunction and ASD-like abnormal behavior remains unclear.

To tackle these issues, we focused on two ASD susceptibility genes, *contactin associated protein-like 2* (*CNTNAP2*) and *Abelson helper integration site-1* (*AHI1*), and examined their roles in synaptic function and behavior. Mutation in *CNTNAP2* gene was identified in Old Order Amish children with cortical dysplasia-focal epilepsy (CDFE) characterized by epileptic seizures, language regression, intellectual disability, hyperactivity and ASD[21]. Subsequent studies show that *CNTNAP2* polymorphisms are a risk factor for ASD[22], and that knockout or knockdown of *Cntnap2* in mice causes ASD-like behavior[19,23], abnormal neural network activity[23], and abnormal synapse maturation and function[19,24–26]. However, there is a contradictory report showing no social deficits in *Cntnap2*-knockout rats[27]. On the other hand, a mutation in *AHI1* gene is one of the causes of Joubert syndrome characterized by agenesis of the cerebellar vermis, cognitive impairment, and delayed development[28,29]. About 30% of patients with Joubert syndrome exhibit ASD[30] and conversely, common genetic variation in *AHI1* gene is reported in patients with ASD[31], suggesting that disruption of *AHI1* leads to symptoms of ASD. Knockout or heterozygous knockout of *Ahi1* in mice is reported to cause depressive phenotypes[32], decreased anxiety, and increased social interaction[33].

In the present study, we investigated whether the loss-of-function of CNTNAP2 or AHI1 resulted in common phenotypes of synaptic dysfunction and ASD-related behavior in mice. We performed knockdown of CNTNAP2 or AHI1 in layer 2/3 pyramidal neurons of the mouse prefrontal cortex (PFC), which has been implicated in social behavior and ASD[23,34–43]. We found that knockdown of CNTNAP2 in layer 2/3 pyramidal neurons of the PFC reduced excitatory and inhibitory synaptic transmission presumably by decreasing the number of functional synapses, elevated inhibition/excitation (I/E) balance, impaired social interaction, and induced mild pup vocalization abnormality. On the other hand, knockdown of AHI1 reduced excitatory synaptic transmission presumably by decreasing postsynaptic AMPA receptor function or its number, elevated I/E balance, impaired social interaction, and induced mild pup vocalization abnormality. Although the causes of reduced excitatory transmission were different, CX546, a positive allosteric modulator of AMPA receptor, effectively enhanced excitatory synaptic transmission, normalized I/E balance, and improved social interaction in both CNTNAP2-knockdown and AHI1-knockdown mice. These results suggest that reduced excitatory synaptic transmission and elevated I/E balance in layer 2/3 pyramidal neurons of the PFC lead to ASD-like behavioral abnormalities in mice.

## Results

**CNTNAP2 knockdown reduces synaptic function in the PFC.** Previous studies show that *Cntnap2*-knockout mice exhibit ASD-like behavioral abnormalities, abnormal neural network activity[23] and synaptic dysfunction[19] in layer 2/3 neurons of the PFC. Indeed, our fluorescent in situ hybridization (FISH) analysis revealed that *Cntnap2* mRNA was expressed not only in layer 2/3 pyramidal neurons of the PFC (Fig. 1a) but also in other cortical layers (Supplementary Fig. 1). To test whether the lack of CNTNAP2 in these neurons is responsible for the phenotypes of the knockout mice, we performed microRNA–mediated knockdown of *Cntnap2* gene specifically in layer 2/3 pyramidal neurons of the mouse PFC during postnatal development. By using in utero electroporation[44], we transfected plasmids carrying micro-RNAs against CNTNAP2 and cDNA for EGFP in pyramidal neurons of the PFC at embryonic day 14–15. We confirmed that microRNAs against CNTNAP2 significantly reduced the expression of CNTNAP2 in human embryonic kidney (HEK) cells (Supplementary Fig. 2a, b). We also verified that transfected cells at postnatal 8–18 weeks were layer 2/3 pyramidal neurons and were mainly located in the PFC, including the dorsolateral prefrontal cortex (dl-PFC), medial prefrontal cortex (mPFC), orbitofrontal cortex (OFC), and anterior cingulate cortex (cgCX) (Supplementary Fig. 3). There were few transfected cells in layer I, V, and VI of the PFC (Supplementary Fig. 4). To estimate how much percentage of layer 2/3 pyramidal neurons were transfected, we performed double immunostaining of the PFC for EGFP (a transfection marker) and CaMKII (a marker for pyramidal neurons). About 25% of CaMKII-positive layer 2/3 pyramidal neurons expressed EGFP in the PFC (Supplementary Fig. 5b, d) and almost all EGFP expressing neurons were CaMKII-positive pyramidal neurons (Supplementary Fig. 5c).

To evaluate the effects of CNTNAP2 knockdown on synapse function in the mPFC, we made whole-cell patch-clamp recording from layer 2/3 pyramidal neurons and compared synaptic responses between EGFP-positive neurons with CNTNAP2 knockdown and EGFP-negative control neurons from P16 to P24 (Fig. 1b). We found that the amplitude of evoked excitatory postsynaptic current (EPSC) of CNTNAP2-knockdown neurons was significantly smaller than that of control neurons (Fig. 1c, d). Furthermore, the frequency but not the amplitude of miniature EPSC (mEPSC) was significantly lower in CNTNAP2-knockdown neurons (Fig. 1e–g), indicating that CNTNAP2 knockdown attenuates excitatory synaptic transmission. No difference was found in the paired-pulse ratio and the NMDA/AMPA ratio (Supplementary Fig. 6a–d). These results suggest that CNTNAP2 knockdown in layer 2/3 pyramidal neurons reduced the number of functional excitatory synapses on these neurons. We then examined the effects of CNTNAP2 knockdown on inhibitory synaptic transmission in layer 2/3 pyramidal neurons. We found that the amplitude of inhibitory postsynaptic current (IPSC) of CNTNAP2-knockdown neurons was significantly smaller than that of control neurons (Fig. 1h, i). The frequency but not the amplitude of miniature IPSC (mIPSC) was significantly lower in CNTNAP2-knockdown neurons (Fig. 1j–l), indicating that CNTNAP2 knockdown reduced inhibitory synaptic transmission. While both EPSCs and IPSCs were reduced, the inhibition/excitation (I/E) ratio was significantly larger in CNTNAP2-knockdown neurons than in control neurons (Fig. 1m, n). We also found that both EPSCs and IPSCs were reduced in CNTNAP2-knockdown neurons in 8 to 11 week-old mice, indicating that the impairment of synaptic function persists into adulthood (Supplementary Fig. 6e–l). The effects of CNTNAP2 knockdown were rescued by co-injection of constructs for the expression of an RNAi-resistant CNTNAP2 (CNTNAP2 rescue) (Fig. 1c–n). Taken together, these results

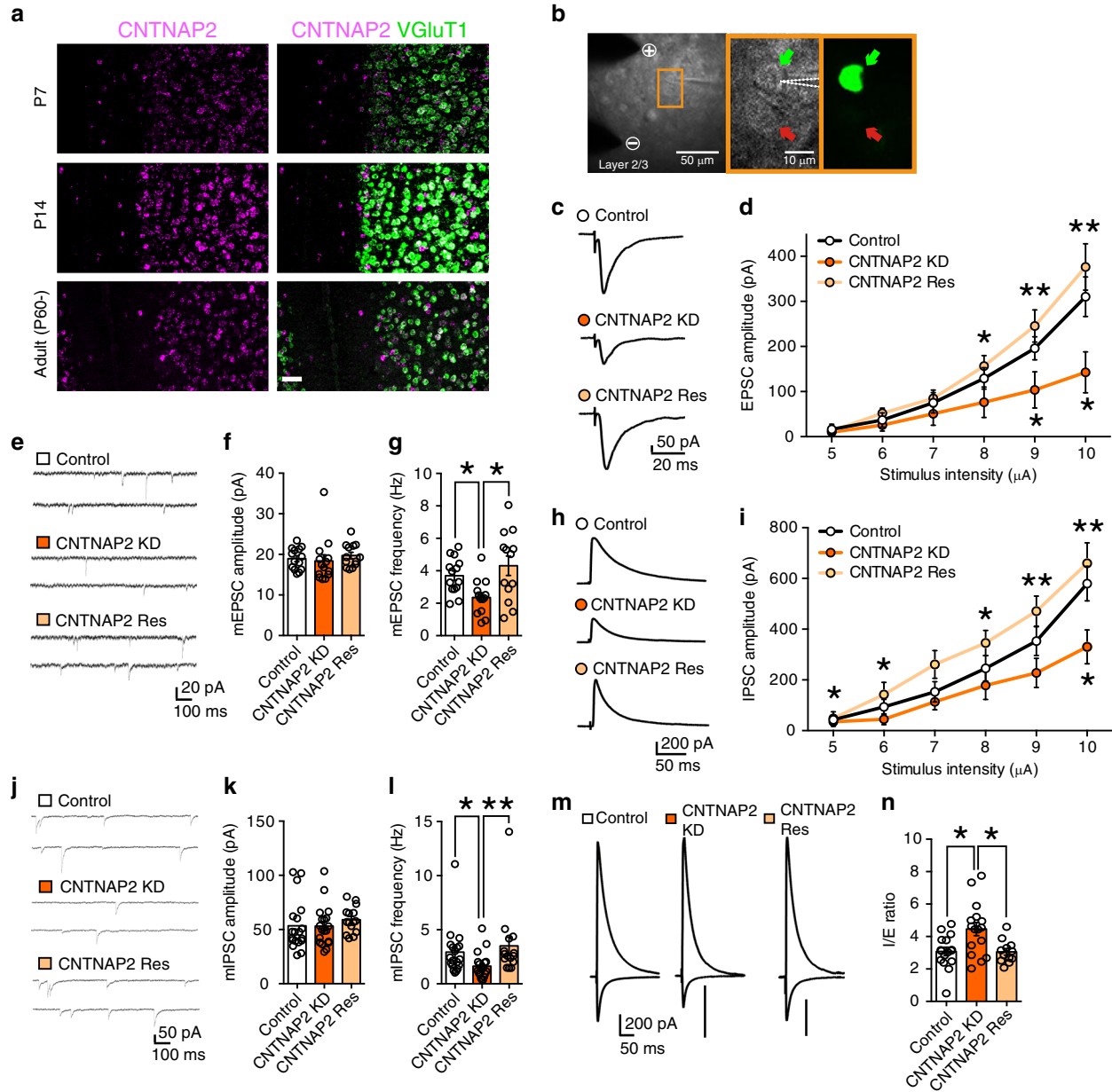

**Fig. 1 Effects of CNTNAP2 knockdown in layer 2/3 pyramidal neurons of the PFC on synaptic function. a** Images from double fluorescence in situ hybridization for mRNA of CNTNAP2 and that of VGluT1 in layer 2/3 pyramidal neurons. Scale bar, 50 μm. **b** Images of a recorded pyramidal neuron. Dashed white lines delineate the patch pipette. + and − show the anode and cathode, respectively, of the stimulation electrode. Bright field (middle) and fluorescent (right) images correspond to the orange rectangle in the left panel. Green and red arrows represent an EGFP-positive knockdown cell and an EGFP-negative control cell, respectively. **c, d** Traces (**c**) and input-output relationships (**d**) for EPSCs in control (white circles, n = 13 cells from 4 mice), CNTNAP2-knockdown (CNTNAP2-KD) (orange circles, n = 13 cells/4 mice) and CNTNAP2-rescue (CNTNAP2-Res) (light orange circles, n = 11 cells/4 mice) cells. **e-g** Traces of mEPSCs (**e**) and summary graphs showing the amplitude (**f**) and frequency (**g**) of mEPSCs for control (white columns, n = 14 cells/4 mice), CNTNAP2-KD (orange columns, n = 14 cells/4 mice) and CNTNAP2-Res (light orange columns, n = 13 cells/4 mice) cells. **h, i** Traces (**h**) and input-output relationships (**i**) for IPSCs in control (white circles, n = 17 cells/3 mice), CNTNAP2-KD (orange circles, n = 16 cells/3 mice) and CNTNAP2-Res (light orange circles, n = 13 cells/2 mice) cells. **j-l** Traces of mIPSCs (**j**) and summary graphs showing the amplitude (**k**), and frequency (**l**) of mIPSCs for control (white columns, n = 19 cells/6 mice), CNTNAP2-KD (orange columns, n = 19 cells/6 mice) and CNTNAP2-Res (light orange columns, n = 15 cells/3 mice) cells. **m, n** Traces (**m**) and the I/E ratio (**n**) for control (white column, n = 16 cells/5 mice), CNTNAP2-KD (orange column, n = 16 cells/5 mice) and CNTNAP2-Res (light orange column, n = 12 cells/3 mice) cells. *$P < 0.05$, **$P < 0.01$ (Dunn test). Data are mean ± SEM. Source data are provided as a Source Data file.

indicate that CNTNAP2 is required for normal excitatory and inhibitory synaptic function in layer 2/3 pyramidal neurons of the PFC, which is consistent with the previous report of reduced excitatory and inhibitory synaptic inputs to cultured cortical neurons by CNTNAP2 knockdown[24].

**ASD-like behavior caused by CNTNAP2 knockdown in the PFC.** We next tested whether CNTNAP2 knockdown in layer 2/3 pyramidal neurons of the developing mouse PFC caused ASD-like behaviors. We first analyzed social interaction using sociality test, three-chamber sociality test, and reciprocal social interaction

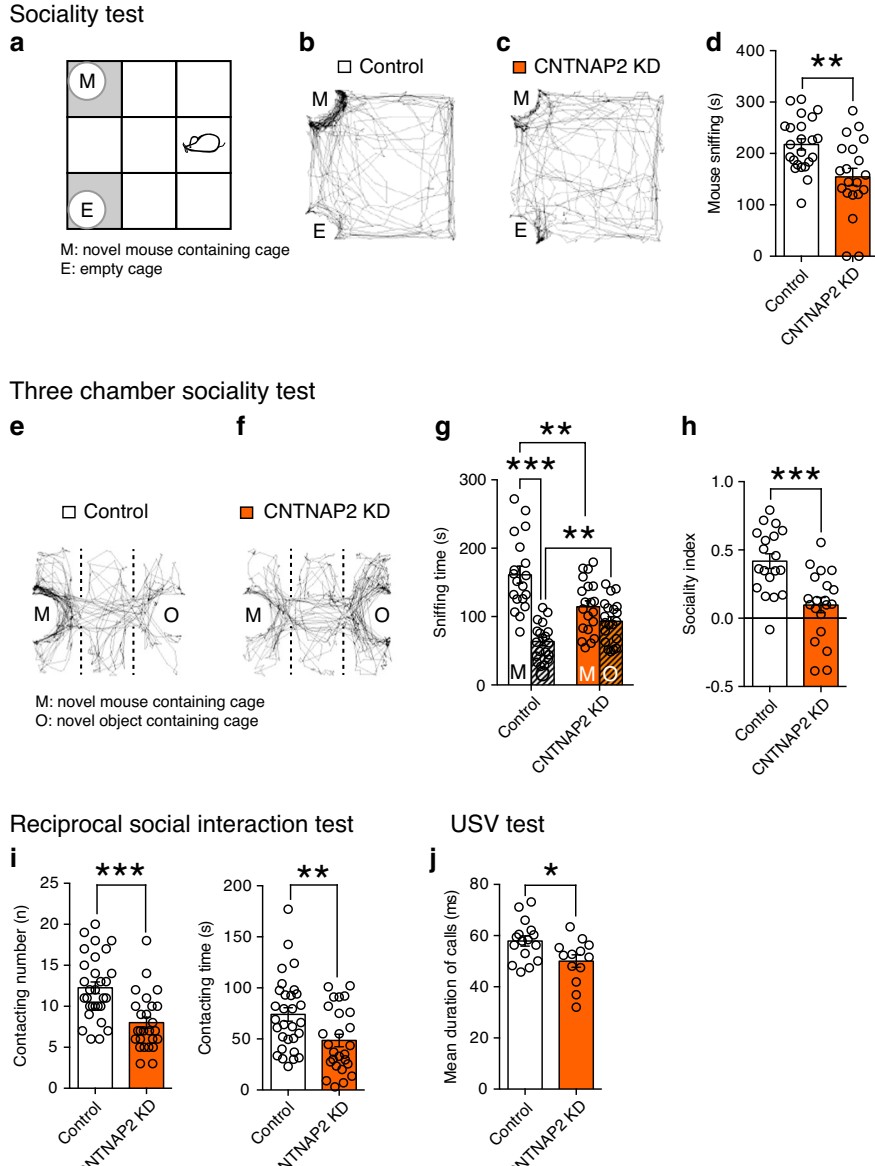

**Fig. 2 Effects of CNTNAP2 knockdown in layer 2/3 pyramidal neurons of the PFC on ASD-like behaviors. a–d** Schema of the sociality test (**a**). Representative tracks (**b**, **c**), and summary graphs showing the amount of time sniffing the novel mouse (M) (**d**) for CNTNAP2-scramble (control) (white columns, $n = 23$) and CNTNAP2-KD (orange columns, $n = 20$) mice. **e–h** Representative tracks (**e**, **f**) and summary graphs showing the amount of time sniffing the novel mouse (M) and the novel object (O) (**g**), and sociality index (**h**) for control (white columns, $n = 19$) and CNTNAP2-KD (orange columns, $n = 20$) mice. **i** Contacting number and time for control (white column, $n = 30$) and CNTNAP2-KD (orange column, $n = 26$) mice. **j** Duration of calls for control (white column, $n = 16$) and CNTNAP2-KD (orange column, $n = 14$) pups. *$P < 0.05$, **$P < 0.01$, ***$P < 0.001$ (Student's $t$ test or Paired $t$ test). Data are mean ± SEM. Source data are provided as a Source Data file.

test. We found that mice with CNTNAP2 knockdown in the PFC displayed a significant decrease in sniffing and stay time around a novel mouse cage in the sociality test (Fig. 2a–d and Supplementary Fig. 8a), which is similar to the phenotype of *Cntnap2*-knockout mice reported previously[23]. There was no difference between control and CNTNAP2-knockdown mice in exploration time for a novel object (Supplementary Fig. 7). In the three-chamber sociality test, CNTNAP2-knockdown mice did not show preference to a novel mouse over a novel object whereas control mice exhibited a clear preference to a novel mouse over a novel object (Fig. 2e–g and Supplementary Fig. 8b). Additionally, the sniffing time to a novel mouse (Fig. 2g) and the sociality index (Fig. 2h) was significantly lower in CNTNAP2-knockdown mice than in control mice. Moreover, in the reciprocal social

interaction test, CNTNAP2-knockdown mice exhibited a significant reduction in the number of contacts and the contact time, and also the number of following, nose-to-nose sniffing, and nose-to-anogenital sniffing, with a freely moving novel mouse (Fig. 2i and Supplementary Fig. 8c). There was no change in anxiety-related behaviors (Supplementary Fig. 9d, e), suggesting that the impairments of social interaction are not likely due to increased anxiety. These data demonstrate that mice with CNTNAP2 knockdown in layer 2/3 pyramidal neurons of the developing PFC are impaired in social interaction.

We next examined whether social vocalization was affected in CNTNAP2-knockdown mice. Mouse pups use ultrasonic vocalization (USV) to communicate with their mothers. We analyzed USV at P7 and found that the pups with CNTNAP2-knockdown

exhibited a small but significant reduction in the mean duration of call (Fig. 2j). This data suggests that pups with CNTNAP2 knockdown display a mild abnormality in vocalization.

To test repetitive behaviors and behavioral flexibility, we measured self-grooming, marble burying, and operant reversal learning. However, we found no significant changes in these behaviors (Supplementary Fig. 9a–c) in CNTNAP2-knockdown mice. The traits of ASD often overlap with those of other neuropsychiatric disorders such as cognitive deficits, attention-deficit hyperactivity disorder, and depression. To examine whether CNTNAP2-knockdown mice exhibit abnormal behaviors related to these disorders, we performed open field test, Y-maze test, forced swim test, and tail suspension test. Although CNTNAP2-knockdown mice showed elevated depressive trait, they exhibited normal locomotor activity and working memory (Supplementary Fig. 9f–i).

The results so far indicate that CNTNAP2 knockdown in layer 2/3 pyramidal neurons of the developing mouse PFC results in reduced excitatory and inhibitory synaptic inputs to these neurons, increased I/E balance, and impaired sociality and mild pup vocalization abnormality relevant to symptoms of ASD.

**AHI1 knockdown reduces excitatory synaptic input in the PFC.** Next, we examined AHI1, an ASD-related gene with unknown function. To examine whether AHI1 deficiency causes deficits in synapse development and function as well as ASD-like behaviors, we knocked down AHI1 in layer 2/3 pyramidal neurons of the developing mouse PFC. We confirmed that microRNAs against AHI1 significantly reduced the expression of AHI1 in HEK cells (Supplementary Fig. 2c, d). We also verified that the distribution of transfected cells was located in layer 2/3 pyramidal neurons of the PFC (Supplementary Fig. 11). FISH analysis showed that AHI1 mRNA was expressed not only in layer 2/3 pyramidal neurons of the PFC (Fig. 3a) but also in other cortical layers (Supplementary Fig. 10).

We then examined roles of AHI1 in synaptic function in layer 2/3 pyramidal neurons of the mPFC. We found that the amplitude of evoked EPSC of AHI1-knockdown neurons was significantly smaller than that of control neurons (Fig. 3b, c). Furthermore, the NMDA/AMPA ratio was significantly larger in AHI1-knockdown neurons than in control neurons (Fig. 3d, e). The amplitude but not the frequency of mEPSC was significantly smaller in AHI1-knockdown neurons than in control neurons (Fig. 3f–h). There was no difference in paired-pulse ratio (Supplementary Fig. 12a, b). These results indicate that AMPA receptor function or its number at postsynaptic sites was reduced in AHI1-knockdown neurons of the mPFC. For inhibitory synaptic transmission, no significant difference was found in either the amplitude or the frequency of mIPSC (Fig. 3i–k). We also found that the excitatory transmission on layer 2/3 pyramidal neurons with AHI1 knockdown was reduced in adult mice (8–11 week-old), indicating almost the same synaptic phenotypes in young and adult mice (Supplementary Fig. 12c–i). The I/E ratio became significantly larger in AHI1-knockdown neurons (Fig. 3l, m). These effects of AHI1 knockdown were rescued by co-injection of an RNAi-resistant construct for AHI1 (AHI1 rescue) (Fig. 3b–h, l, m). These results strongly suggest that knockdown of AHI1 leads to specific reduction in postsynaptic AMPA receptor function or its number at excitatory postsynaptic sites of layer 2/3 pyramidal neurons of the PFC.

A previous study demonstrated that huntingtin-associated protein 1 (HAP1) interacted with AHI1[45]. We therefore knocked down HAP1 in layer 2/3 pyramidal neurons of the mPFC and examined synaptic transmission. We found that knockdown of HAP1 caused significant reduction in excitatory synaptic

transmission (Fig. 3n, o and Supplementary Figs. 2e, f and 13a–e), which were essentially identical to the phenotypes caused by AHI1 knockdown (Fig. 3b–m). Furthermore, double-knockdown of AHI1 and HAP1 had the same effect as single knockdown of AHI1 (Fig. 3p, q and Supplementary Fig. 13f–j), indicating that AHI1 regulates synaptic function through HAP1.

**ASD-like behavior caused by AHI1 knockdown in the PFC.** To test whether mice with AHI1 knockdown exhibit ASD-like behaviors, we investigated social interaction, vocalization, repetitive behaviors, and behavioral flexibility. In the sociality test, AHI1-knockdown mice spent significantly less time around the cage of a novel mouse than control mice expressing AHI1-scramble RNA in layer 2/3 pyramidal neurons of the PFC (Fig. 4a–c and Supplementary Fig. 15a). There was no difference between control and AHI1-knockdown mice in exploration time for a novel object (Supplementary Fig. 14). In the three-chamber sociality test, both AHI1-knockdown mice and control mice showed preference to a novel mouse over a novel object (Fig. 4d–f), indicating that AHI1-knockdown mice had sociality. However, the sniffing time to a novel mouse (Fig. 4f), stay time in the mouse chamber (Supplementary Fig. 15b) and the sociality index (Fig. 4g) was significantly lower in AHI1-knockdown mice than in control mice. Moreover, in the reciprocal social interaction test, AHI1-knockdown mice exhibited a significantly smaller number of contact and nose-to-nose sniffing as well as significantly shorter contacting time with a freely moving novel mouse than control mice (Fig. 4h and Supplementary Fig. 15c). In the USV analysis at P7, AHI1-knockdown pups exhibited a small but significant reduction in the mean duration of call (Fig. 4i). However, neither repetitive behavior nor behavioral inflexibility was altered in AHI1-knockdown mice. These results suggest that AHI1-knockdown mice display reduced social interaction and mild pup vocalization abnormality.

To examine whether AHI1 knockdown in the PFC caused other behavioral abnormalities, we performed a battery of behavioral tests. AHI1 knockdown caused decreased depressive behavior, but did not alter exploratory behavior, anxiety, locomotor activity, and working memory (Supplementary Figs. 14 and 16).

**CX546 ameliorates ASD-like behaviors of the knockdown mice.** The results presented so far suggest that reduced excitatory synaptic transmission in layer 2/3 pyramidal neurons of the PFC may be responsible for ASD-like behaviors in CNTNAP2-knockdown and AHI1-knockdown mice. To test this possibility, we used the ampakine CX546, a positive allosteric modulator of AMPA receptors[20]. First, we evaluated the effect of CX546 on the excitatory synaptic transmission in layer 2/3 pyramidal neurons with CNTNAP2 knockdown or AHI1 knockdown. We found that the decay time and charge transfer of mEPSC became significantly longer and larger, respectively, after CX546 treatment in control, CNTNAP2-knockdown, and AHI1-knockdown neurons (Fig. 5a–c, i–k). In contrast, there was no difference in the amplitude and frequency of mIPSC between CX546-untreated and -treated neurons for control, CNTNAP2-knockdown, and AHI1-knockdown neurons (Fig. 5d–f, l–n). These results indicate that CX546 effectively enhanced excitatory synaptic transmission without affecting inhibitory synaptic transmission. As for the effect of CX546 on I/E ratio, we found no difference between CX546-untreated and -treated neurons in control neurons (Fig. 5g, h, o, p). In contrast, CX546 treatment significantly decreased the I/E ratio in CNTNAP2- or AHI1-knockdown neurons (Fig. 5g, h, o, p), indicating that CX546 normalizes I/E balance.

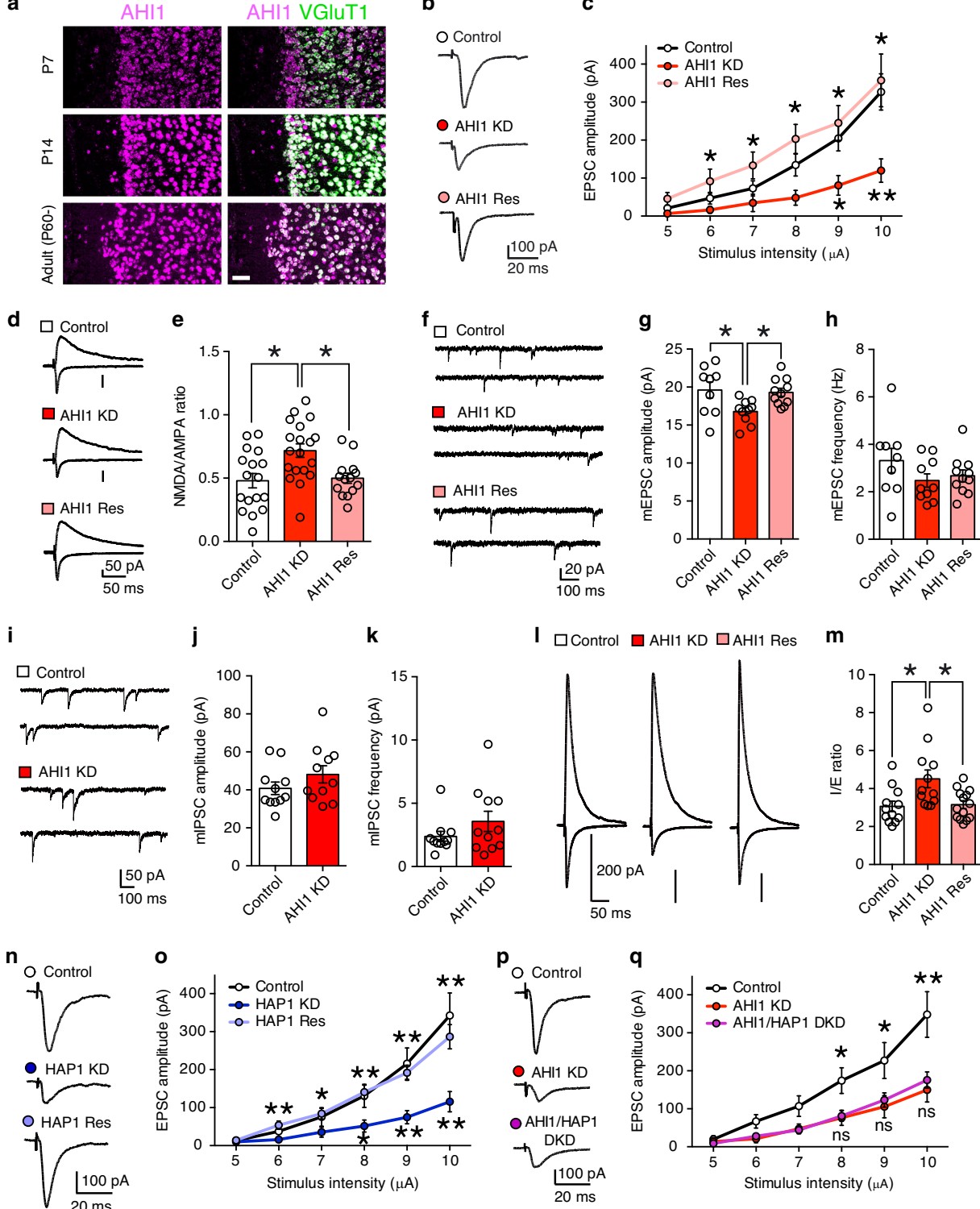

We then examined whether CX546 amended the ASD-like abnormal behaviors seen in CNTNAP2-knockdown and AHI1-knockdown mice. We found that the injection of CX546 significantly improved their scores in both sociality and reciprocal social interaction tests (Fig. 6 and Supplementary Fig. 17). In contrast, CX546 had no significant effects on social interaction in CNTNAP2-scramble and AHI1-scramble mice (Fig. 6 and Supplementary Fig. 17) despite a clear increase in excitatory synaptic transmission in control neurons after CX546

treatment (Fig. 5a–c, i–k). This result is consistent with a previous study that social interaction was normal in C57B6 mice after injection of various ampakine compounds[20]. One possibility for the ineffectiveness of ampakines would be that control mice may exhibit maximum performance of social interaction under normal level of excitatory synaptic transmission. Another possibility would be that I/E balance, which was not altered by CX546 in control neurons (Fig. 5g, h, o, p), may be a critical factor for social interaction.

**Fig. 3 AHI1 regulates excitatory synaptic transmission through HAP1. a** Images from double fluorescence in situ hybridization for mRNA of AHI1 and that of VGluT1 in layer 2/3 pyramidal neurons. Scale bar, 50 μm. **b**, **c** Representative traces (**b**) and input-output relationships (**c**) for EPSCs in control (white circles, *n* = 11 cells/2 mice), AHI1-knockdown (AHI1-KD) (red circles, *n* = 10 cells/2 mice) and AHI1-rescue (AHI1-Res) (light red circles, *n* = 11 cells/4 mice) cells. **d**, **e** Traces (**d**) and NMDA/AMPA ratio (**e**) for control (white column, *n* = 17 cells/6 mice), AHI1-KD, (red column, *n* = 19 cells/7 mice) and AHI1-Res (light red column, *n* = 14 cells/5 mice) cells. **f–h** Traces of mEPSC (**f**) and summary graphs showing the amplitude (**g**) and frequency (**h**) of mEPSC for control (white columns, *n* = 9 cells/3 mice), AHI1-KD (red columns, *n* = 10 cells/3 mice) and AHI1-Res (light red columns, *n* = 11 cells/3 mice) cells. **i–k** Traces of mIPSC (**i**) and summary graphs showing the amplitude (**j**) and frequency (**k**) of mIPSC for control (white columns, n = 11 cells/3 mice) and AHI1-KD (red columns, *n* = 11 cells/3 mice) cells. **l**, **m** Traces (**l**) and the I/E ratio (**m**) for control (white column, *n* = 12 cells/3 mice), AHI1-KD (orange column, *n* = 12 cells/3 mice) and AHI1-Res (light orange column, *n* = 14 cells/2 mice) cells. **n**, **o** Traces (**n**) and input-output relationships (**o**) for EPSCs in control (white circles, *n* = 13 cells/5 mice), HAP1-knockdown (HAP1-KD) (blue circles, *n* = 14 cells/5 mice) and HAP1-rescue (HAP1-Res) (light blue circles, *n* = 13 cells/5 mice) cells. **p**, **q** Traces (**p**) and input-output relationships (**q**) for EPSCs in control (white circles, *n* = 12 cells/4 mice), AHI1-KD (red circles, *n* = 11 cells/4 mice) and AHI1/HAP1 double-knockdown (AHI1/HAP1-DKD) (purple circles, *n* = 16 cells/3 mice) cells. *P < 0.05, **P < 0.01 (Dunn test). Data are mean ± SEM. Source data are provided as a Source Data file.

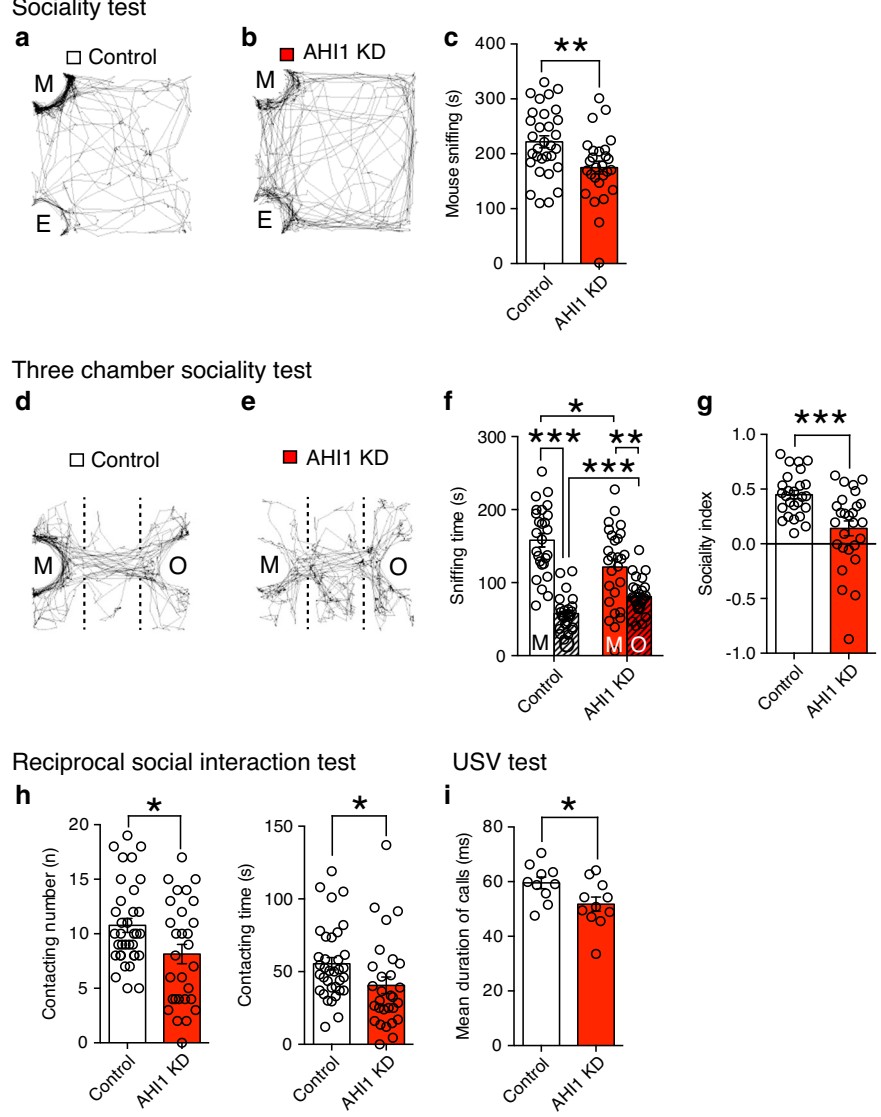

**Fig. 4 Effects of AHI1 knockdown in layer 2/3 pyramidal neurons of the PFC on ASD-like behaviors. a–c** Representative tracks (**a**, **b**) and summary graphs showing the amount of time sniffing the novel mouse (M) (**c**) for AHI1-scramble (control) (white columns, *n* = 31) and AHI1-KD (red columns, *n* = 27) mice. **d–g** Representative tracks (**d**, **e**) and summary graphs showing the amount of time sniffing the novel mouse (M) and the novel object (O) (**f**) and sociality index (**g**) for control (white columns, *n* = 26) and AHI1-KD (orange columns, *n* = 27) mice. **h** Contacting number and time in control (white column, *n* = 35) and AHI1-KD (red column, *n* = 29) mice. **i** Duration of calls in control (white column, *n* = 10) and AHI1-KD (red column, *n* = 11) pups. *P < 0.05, ***P < 0.001 (Student's *t* test or Paired *t* test). Data are mean ± SEM. Source data are provided as a Source Data file.

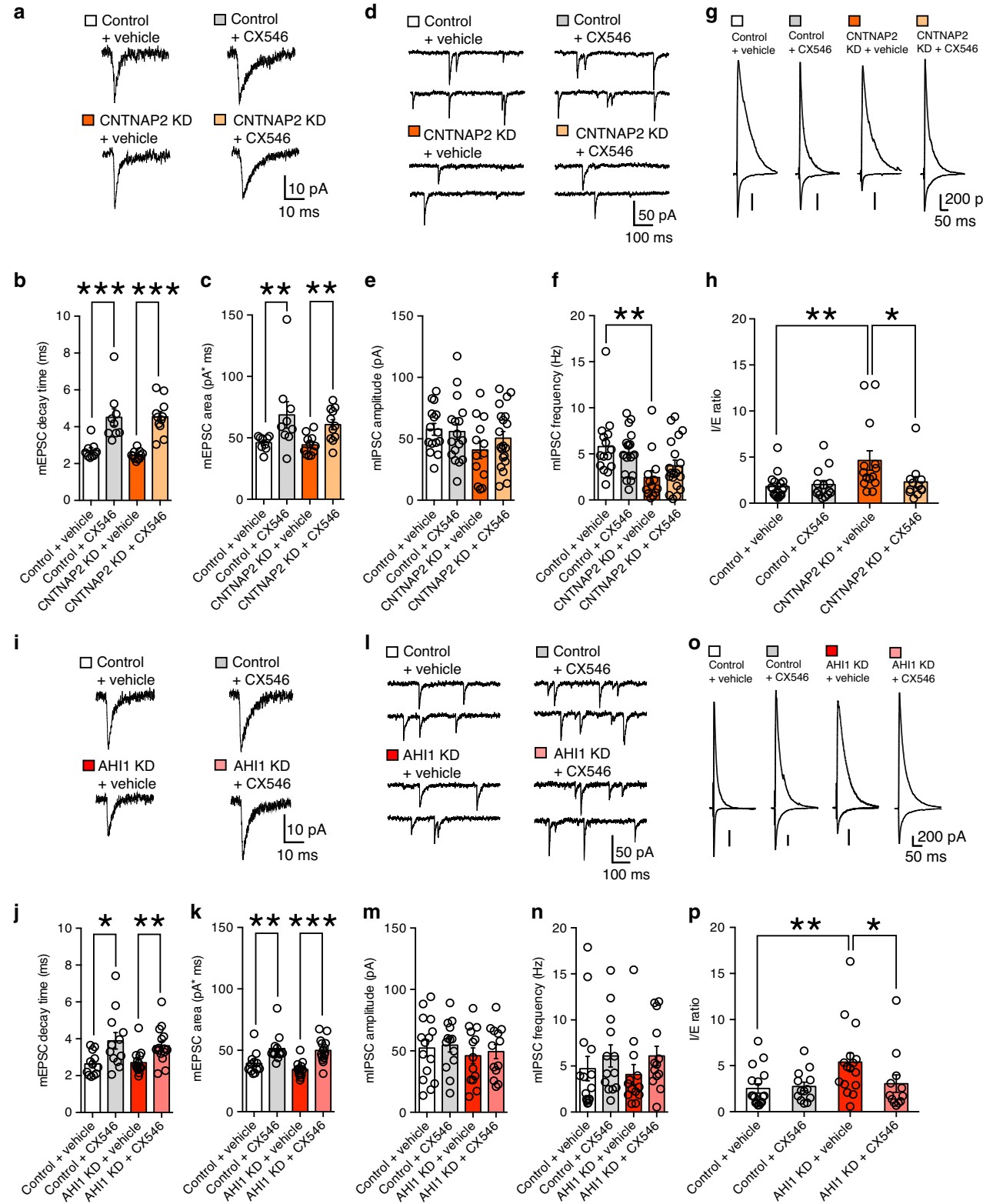

Since CX546 was administered systematically, it should affect excitatory synapses in the whole brain. We, therefore, assume that CX546 may enhance excitatory synaptic transmission in brain regions other than the mPFC. However, the above results that CX546 had no effect on social interaction in control mice suggest that reduced excitatory synaptic transmission in layer 2/3 pyramidal neurons of the PFC is the main cause of impaired social interaction in CNTNAP2-knockdown and AHI1-knockdown mice. In addition, the present data may provide a basis for potential therapeutic use of ampakines for impaired social interaction, as reported previously[19,20].

## Discussion

Recent studies have identified numerous genes that have mutations in patients with ASD[1,2]. Many ASD mouse models have

**Fig. 5 CX546 enhances excitatory synaptic transmission in CNTNAP2-knockdown and AHI1-knockdown mice. a–c** Traces of mEPSCs (**a**) and graphs showing the decay time constant (**b**) and the charge transfer (**c**) of mEPSCs for vehicle-treated control (white columns, $n = 9$ cells/6 mice), CX546-treated control (gray columns, $n = 9$ cells/6 mice), vehicle-treated CNTNAP2-KD (orange columns, $n = 11$ cells/7 mice) and CX546-treated CNTNAP2-KD (light orange columns, $n = 11$ cells/7 mice) cells. **d–f** Traces of mIPSCs (**d**), and graphs showing the amplitude (**e**) and frequency (**f**) of mIPSCs for vehicle-treated control (white columns, $n = 15$ cells/3 mice), CX546-treated control (gray columns, $n = 17$ cells/3 mice), vehicle-treated CNTNAP2-KD (orange columns, $n = 13$ cells/3 mice) and CX546-treated CNTNAP2-KD (light orange columns, $n = 20$ cells/3 mice) cells. **g–h** Traces (**g**) and the I/E ratio (**h**) for vehicle-treated control (white columns, $n = 17$ cells/3 mice), CX546-treated control (gray columns, $n = 14$ cells/3 mice), vehicle-treated CNTNAP2-KD (orange columns, $n = 14$ cells/3 mice) and CX546-treated CNTNAP2-KD (light orange columns, $n = 12$ cells/3 mice) cells. **i–k** Similar to **a–c** but results of AHI1 KD, for vehicle-treated control (white columns, $n = 12$ cells/6 mice), CX546-treated control (gray columns, $n = 12$ cells/6 mice), vehicle-treated AHI1-KD (red columns, $n = 15$ cells/6 mice) and CX546-treated AHI1-KD (light red columns, $n = 15$ cells/6 mice) cells. **l–n** Similar to **d–f** but results of AHI1 KD, for vehicle-treated control (white columns, $n = 15$ cells/3 mice), CX546-treated control (gray columns, $n = 13$ cells/3 mice), vehicle-treated AHI1-KD (red columns, $n = 13$ cells/3 mice) and CX546-treated AHI1-KD (light red columns, $n = 13$ cells/3 mice) cells. **o–p** Similar to **g–f** but results of AHI1 KD, for vehicle-treated control (white columns, $n = 17$ cells/3 mice), CX546-treated control (gray columns, $n = 13$ cells/3 mice), vehicle-treated AHI1-KD (red columns, $n = 16$ cells/3 mice) and CX546-treated AHI1-KD (light red columns, $n = 12$ cells/3 mice) cells. *$P < 0.05$, **$P < 0.01$, ***$P < 0.001$ (Mann–Whitney $U$-test). Data are mean ± SEM. Source data are provided as a Source Data file.

been generated by genetic manipulation to study molecular and cellular mechanisms underlying ASD[10,12,13,15,16,40,41,46]. The results from the analyses of these ASD mouse models suggest possible causes of ASD including aberrant mRNA translation, abnormal NMDA receptor function, and imbalance between excitation and inhibition[17,47,48]. However, roles of many ASD-related genes in synapse development and function, their contributions to ASD-like behaviors, and brain regions where they function remain largely unknown. To tackle these issues, we employed in utero electroporation to knockdown ASD-related genes specifically in layer 2/3 pyramidal neurons of the PFC during development, which was used to examine the roles of *Disrupted-in-Schizophrenia-1 (DISC1)* gene in synaptic function and schizophrenia-related behaviors[49,50]. We confirmed that these methods are also useful for investigating roles of ASD-associated genes in synaptic development and function in the PFC, and ASD-like behaviors. Thus, this assay system enables us to characterize ASD-related genes of unknown function and to facilitate the elucidation of mechanisms underlying ASD-like phenotypes.

Brain regions responsible for ASD-like behaviors have been explored in several previous studies. For example, optogenetic activation of neurons in the mPFC induced deficits in social interaction[43]. Disturbance of neuroligin-3 function in striatal medium spiny neurons expressing D1 dopamine receptors enhanced repetitive behaviors[51]. Deletion of Tsc1 in cerebellar Purkinje cells led to ASD-like behaviors including impaired social interaction and communication, and stereotyped and repetitive behaviors[18]. In the present study, we found that knockdown of CNTNAP2 or AHI1 in layer 2/3 pyramidal neurons of the mouse PFC reduced excitatory synaptic transmission and caused the reduction of social interaction and the mild vocalization abnormality. While CNTNAP2 knockdown presumably reduced the number of functional excitatory synapses and also that of inhibitory synapses, AHI1 knockdown presumably reduced postsynaptic AMPA receptor function or its number without affecting inhibitory synaptic transmission. Nevertheless, knockdown of either CNTNAP2 or AHI1 similarly elevated I/E ratio of synaptic responses in layer 2/3 pyramidal neurons of the mPFC. Moreover, despite the different causes for the reduced excitatory transmission, CX546 enhanced excitatory synaptic transmission, normalized I/E balance, and rescued the reduced social interaction in both CNTNAP2-knockdown and AHI1-knockdown mice. These results unequivocally indicate that proper strength of excitatory synaptic transmission and proper I/E balance of synaptic inputs to layer 2/3 pyramidal neurons of the PFC are essential for social interaction and communication. We assume that the reduced inhibitory transmission in CNTNAP2

knockdown is not compensatory for the reduced excitation because AHI1-knockdown mice with reduced excitatory transmission have normal inhibitory transmission. We also speculate that reduced inhibitory transmission is not causal for ASD-like behaviors because CNTNAP2-knockdown and AHI1-knockdown mice exhibit very similar behavioral abnormalities. Taken together with the results of the previous and present studies, ASD-like behaviors appear to be caused by abnormalities of multiple brain regions. How these brain areas or neural circuits interact to regulate ASD-related behaviors awaits further investigation.

The elevated I/E ratio in our CNTNAP2-knockdown and AHI1-knockdown mice might have caused reduction of intrinsic neuronal excitability in the PFC. Indeed, several previous studies show changes in both synaptic inputs and intrinsic neuronal excitability[52–55]. For example, in *Cntnap2*-knockout mice, layer 2/3 excitatory neurons of the somatosensory cortex are reported to have reduced synaptic inputs and increased intrinsic excitability[52]. However, inconsistent results have been reported regarding changes of synaptic inputs and those of intrinsic neuronal excitability in several ASD models[52,56–58]. Therefore, alteration of synaptic inputs may not necessarily accompany change in intrinsic neuronal excitability in ASD mouse models.

The USV test has been currently used to evaluate social communication between pups and their mother. We found that pups with CNTNAP2 knockdown or AHI1 knockdown displayed abnormal vocalization. However, the communicative role of pups-emitted vocalizations remains to be determined because abnormal vocalization is not consistently observed in infants with autism (abnormal infant crying is a risk in autism, but not communication deficits)[59]. Although vocalizations in juvenile and adult mice might provide better indices of communication, methods for measuring such communication have not been established[60]. Social odor assay might be a useful means to investigate communication in mouse models of ASD[59], which should be adopted to investigate sociality of our ASD mouse models in future studies.

We found no change in repetitive behaviors and behavioral flexibility in mice with CNTNAP2 knockdown or AHI1 knockdown. These results suggest that excitatory synaptic transmission of layer 2/3 pyramidal neurons in the PFC may not be involved in repetitive behaviors and behavioral flexibility. However, considering that only ~25% of layer 2/3 pyramidal neurons were transfected with CNTNAP2-knockdown vectors in the present experimental condition (Supplementary Fig. 5b), it is possible that the proportion of knockdown neurons was too small to cause discernible deficits in repetitive behaviors and behavioral inflexibility.

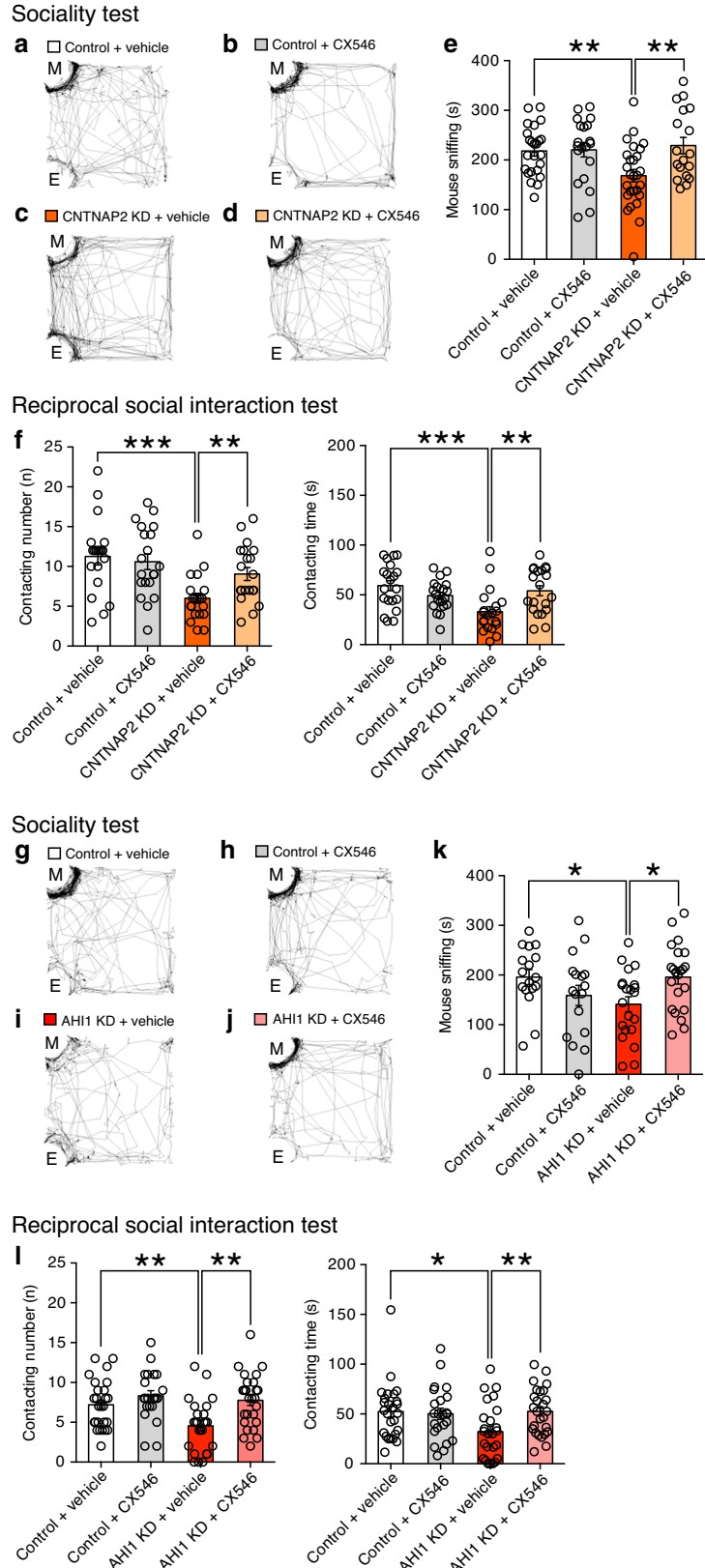

It has been shown that AHI1 is mutated in patients with Joubert syndrome and its deletion leads to cerebellar hypoplasia presumably by a proliferation deficit[28,61]. In the present study, we have disclosed a new role of AHI1 as a regulator of excitatory synaptic function presumably through enhancing postsynaptic AMPA receptor function or its expression. Previous studies show that AHI1 interacts with huntingtin-associated protein 1 (HAP1) and the deletion of AHI1 results in reduced HAP1 protein in the mouse brain[32,45]. In addition, HAP1 is suggested to be involved in AMPA receptor trafficking and its function through the kinesin motor protein 5 that contributes to the transport of AMPA receptors[62]. In the present study, we demonstrated that

**Fig. 6 CX546 improves reduced social interaction in CNTNAP2-knockdown and AHI1-knockdown mice. a–e** Representative tracks (**a–d**) and summary graphs showing the amount of time sniffing the mouse (M) (**e**) in vehicle-treated control (white column, $n = 24$), CX546-treated control (gray column, $n = 20$), vehicle-treated CNTNAP2-KD (orange column, $n = 28$) and CX546-treated CNTNAP2-KD (light orange column, $n = 20$) mice. **f** Summary graphs showing the contacting number and time in vehicle-treated control (white column, $n = 20$), CX546-treated control (gray column, $n = 20$ mice), vehicle-treated CNTNAP2-KD (orange column, $n = 20$) and CX546-treated CNTNAP2-KD (light orange column, $n = 20$) mice. **g–k** Representative tracks (**g–j**) and summary graphs showing the amount of time sniffing the mouse (M) (**k**) in vehicle-treated control (white column, $n = 17$), CX546-treated control (gray column, $n = 17$), vehicle-treated AHI1-KD (red column, $n = 21$) and CX546-treated AHI1-KD (light red column, $n = 22$) mice. **l** Summary graphs showing the contacting number and time in vehicle-treated control (white column, $n = 26$), vehicle-treated control (gray column, $n = 23$), vehicle-treated AHI1-KD (red column, $n = 25$) and CX546-treated AHI1-KD (light red column, $n = 26$) mice. *$P < 0.05$, ***$P < 0.001$ (Student's $t$ test). Source data are provided as a Source Data file.

HAP1 knockdown in layer 2/3 pyramidal neurons in the mPFC reduced excitatory synaptic transmission, which was very similar to the phenotype of AHI1 knockdown and was presumably caused by reduced postsynaptic AMPA receptor function or its expression. Furthermore, the effect of HAP1 knockdown was occluded by double-knockdown of AHI1 and HAP1. These results strongly suggest that HAP1 functions downstream of AHI1 and that knockdown of AHI1 disrupts HAP1 function and impairs AMPA receptor trafficking and function.

A previous study shows that social interaction time is significantly increased in AHI1 heterozygous knockout mice[33]. By marked contrast, we demonstrate in the present study that mice with AHI1 knockdown specifically in layer 2/3 pyramidal neurons of the PFC showed reduced social interaction and mild vocalization abnormality. This apparent discrepancy may be ascribed to the difference in the anxiety level between the two mouse models because social behaviors are closely linked to anxiety. While the anxiety level was normal in mice with AHI1 knockdown in layer 2/3 pyramidal neurons of the PFC, it was decreased in AHI1 heterozygous knockout mice[33]. It is therefore likely that the increased social interaction in AHI1 heterozygous knockout mice may result at least partly from the reduced anxiety level.

Diverse synaptic dysfunctions have been reported to contribute to the etiology of ASD[17,47,63]. Notably, opposite synaptic defects are found in different ASD mouse models[10,12–14,16,41,46]. For instance, Shank2-knockout mice (lacking exons 6 and 7) show reduced NMDAR function[15], whereas another line of Shank2-knockout mice (lacking exon 7) and IRSp53-knockout mice show increased NMDAR function[13,46]. Inhibitory synaptic transmission is reported to be increased in neuroligin-3 knock-in R451C mice[14], whereas it was found to be reduced in BTBR mice[64]. Several studies showed increase of I/E ratio[14,56,58,65,66], whereas other studies reported decrease of I/E ratio[16,53,57,67]. Furthermore, a recent report showed that *Cntnap2*-knockout mice exhibited significantly reduced I/E ratio due to larger decrease in inhibition than in excitation for synaptic inputs from layer 4 to layer 2/3 pyramidal neurons of the mouse somatosensory cortex[52]. Therefore, it is possible that the direction of change in I/E balance is variable among different ASD mouse models and also among different brain areas and neuron types. However, these seemingly contradicting results suggest that proper I/E balance is important and deviation of the balance from the normal range may be a risk factor for ASD-like behaviors. Thus, individual ASD patients are thought to have variable changes in excitatory or inhibitory synaptic functions depending on the genes mutated. Therefore, it is important to elucidate the roles of individual ASD-related genes in synaptic function and categorize them based on the types of synaptic function in which they are involved.

In conclusion, we have demonstrated that knockdown of CNTNAP2 and that of AHI1 in layer 2/3 pyramidal neurons of the mouse PFC reduced excitatory synaptic transmission, elevated I/E ratio of synaptic responses, reduced social interaction, and

induced mild pup vocalization abnormality. Although the cause of reduced excitatory transmission was clearly distinct, i.e., reduced synapse number for CNTNAP2 knockdown and reduced postsynaptic AMPA receptor function or its number for AHI1 knockdown, pharmacological enhancement of AMPA receptor function effectively restored the reduced social interaction. These results suggest that irrespective of the causes and underlying molecular mechanisms, reduced excitatory synaptic transmission and elevated I/E ratio in layer 2/3 pyramidal neurons of the PFC leads to reduced social interaction and vocalization abnormality, which may underlie the etiology of a subset of patients with ASD. Thus, enhancement of excitatory synaptic transmission might be effective for the treatment of such ASD patients.

## Methods

**Animals**. ICR mice, DBA2 mice, and C57/B6 (SLC JAPAN) were used in the present study. All experiments were performed in accordance with the guidelines set down by the experimental animal ethics committees of The University of Tokyo, Hokkaido University and The Japan Neuroscience Society.

**Preparation of vector constructs**. Vectors were designed to express EGFP, micro RNA (miRNA), and/or cDNA under the control of the CAG promoter[68]. The cDNA for CNTNAP2 and that for AHI1 were obtained by RT-PCR of a cDNA library from the P1 and P30 mouse cortex. Each cDNA was subcloned into pCAG vectors. The BLOCK-iT Pol II miR RNAi expression vector kit (Invitrogen, CA, USA) was used for vector-based RNA interference (RNAi) analysis. The following engineered microRNAs were designed according to the BLOCK-iT Pol II miR RNAi Expression Vector kit guidelines (Invitrogen):

5′-TGCTGTGATCTAGGTGCCAAGGGTCAGTTTTGGCCACTGACTGACT GACCCTTCACCTAGATCA-3′ and 5′-CCTGTGATCTAGGTGAAGGGTCAG TCAGTCAGTGGCCAAAACTGACCCTTGGCACCTAGATCAC-3′ for CNTNA P2-microRNA-1; 5′-TGCTGTACAAGGTCAATCTCCACATTGTTTTGGCCAC TGACTGACAATGTGGATTGACCTTGTA-3′ and 5′-CCTGTACAAGGTC AATCCACATTGTCAGTCAGTGGCCAAAACAATGTGGAGATTGACCTTGT AC-3′ for CNTNAP2-microRNA-2; 5′-TGCTGATAAGAAGCCAGCATCCTT CCGTTTTGGCCACTGACTGACGGAAGGATTGGCTTCTTAT-3′ and 5′-CC TGATAAGAAGCCAATCCTTCCGTCAGTCAGTGGCCAAAACGGAAGGATG CTGGCTTCTTATC-3′ for CNTNAP2-microRNA-3; 5′-TGCTGTAGTGCTG AAGCTAAAGTGGAGTTTTGGCCACTGACTGACTCCACTTTCTTCAGCA CTA-3′ and 5′-CCTGTAGTGCTGAAGAAAGTGGAGTCAGTCAGTGGCCA AAACTCCACTTTAGCTTCAGCACTAC-3′ for CNTNAP2-microRNA-4; 5′-TGCTGATAAGATGAAACAGGACGTTCGTTTTGGCCACTGACTGACGAAC GTCCTTTCATCTTAT-3′ and 5′-CCTGATAAGATGAAAGGACGTTCGTCA GTCAGTGGCCAAAACGAACGTCCTGTTTCATCTTATC-3′ for AHI1-micro-RNA-1; 5′-TGCTGTTGAATGGCAGGTCAGAGTACGTTTTGGCCACTGACTG ACGTACTCTGCTGCCATTCAA-3′ and 5′-CCTGTTGAATGGCAGCAGAGTA CGTCAGTCAGTGGCCAAAACGTACTCTGACCTGCCATTCAAC-3′ for AHI1-microRNA-2; 5′-TGCTGAACAGATGCGTACTCGAAGCTGTTTTGGCCACTGA CTGACAGCTTCGAACGCATCTGTT-3′ and 5′-CCTGAACAGATGCGTTCGA AGCTGTCAGTCAGTGGCCAAAACAGCTTCGAGTACGCATCTGTTC-3′ for AHI1-microRNA-3. These oligonucleotides were subcloned into a CAG vector. The QuikChange Lightning site-directed mutagenesis kit (Agilent Technologies, CA, USA) was used to generate RNAi-resistant forms of CNTNAP2 and AHI1 (CNTNAP2-rescue and AHI1-rescue) cDNAs, which harbor sense mutations (no alteration of amino acid codons). The CNTNAP2-rescue cDNA was fused to mOrange2. The AHI1-rescue cDNA was linked in-frame to mOrange2 interposed by a picornavirus "self-cleaving" P2A peptide sequence to enable efficient bicistronic expression. The CNTNAP2-rescue and AHI1-rescue cDNAs were subcloned into pCAG vectors. Scrambled control miRNA for CNTNAP2 and AHI1 was designed by shuffling the recognition region sequences. A BLAST search confirmed that the

scrambled sequences had no target gene. All constructs were verified by DNA sequencing.

**Electrophysiological recordings from pyramidal neurons in acute cortical slices.** Coronal slices containing the mPFC (300 μm thick) were prepared from mice from P16 to P24 or from 8 to 11 weeks by using a vibratome slicer (Leica, Germany) in a chilled (0–4 °C) cutting solution containing 120 mM Choline-Cl, 28 mM NaHCO₃, 1.25 mM NaH₂PO₄, 2 mM KCl, 25 mM glucose, 1 mM CaCl₂, 8 mM MgCl₂, bubbled with 95% O₂ and 5% CO₂[69]. The preparations were recovered for at least 30 min at room temperature by incubation in a reservoir chamber bathed in a solution containing 125 mM NaCl, 2.5 mM KCl, 2 mM CaCl₂, 1 mM MgSO₄, 1.25 mM NaH₂PO₄, 26 mM NaHCO₃, and 20 mM glucose bubbled with 95% O₂ and 5% CO₂[70]. The preparations were transferred to a recording chamber located on the stage of an Olympus BX51WI microscope. Whole-cell recordings were made from visually identified or fluorescent protein-positive pyramidal neurons in the mPFC (ACC, PL, and/or IL) using an upright and fluorescence microscope at 32 °C. The resistance of patch pipettes was 2–4 MΩ when filled with an intracellular solution composed of 120 mM CsMeSO₃, 15 mM CsCl, 8 NaCl, 0.2 mM EGTA, 10 mM HEPES, 10 mM TEA-Cl, 4 mM MgATP, 0.3 mM Na₂GTP, 0.1 mM Spermine, 5 mM Qx314 (pH7.3, adjusted with CsOH) for measuring the input-output relationship, paired-pulse ratio, NMDA/AMPA ratio, and I/E ratio[71]. A miniature EPSC (mEPSC) was measured with a pipette solution containing 130 mM K D-gluconate, 6 mM KCl, 10 mM NaCl, 10 mM HEPES, 0.16 mM CaCl₂, 2 mM MgCl₂, 0.5 mM EGTA, 4 mM Na-ATP, and 0.4 mM Na-GTP (pH7.3, adjusted with KOH)[72]. A miniature IPSC (mIPSC) was measured with a pipette solution containing 145 mM KCl, 10 mM HEPES, 5 mM ATP.Mg, 0.2 mM GTP-Na, 0.16 mM CaCl₂, 2 mM MgCl₂, and 10 mM EGTA (pH7.2, adjusted with KOH)[73]. Membrane currents were recorded with an EPC-10 amplifier (HEKA Elektronik, Lambrecht/Pfalz, Germany). The signals were filtered at 2.9 kHz and digitized at 20 kHz. The pipette access resistance was compensated by 70%.

Excitatory postsynaptic currents (EPSCs) were recorded from layer 2/3 pyramidal neurons of the mPFC and stimulation was delivered with a bipolar tungsten stimulating electrode (FHC, Inc, USA) placed in layer 2/3 of the mPFC[74]. The recording chamber was continuously perfused with oxygenated bath solution supplemented with picrotoxin (0.1 mM). The holding potential for recording EPSCs was −70 mV. For measuring the paired-pulse ratio, double stimulation pulses were delivered at varying interpulse intervals of 20 ms, 50 ms, 100 ms, and 500 ms. Five to ten consecutive traces were averaged to obtain a mean EPSC trace. For measuring the NMDA/AMPA ratio of evoked EPSCs, AMPA receptor-mediated and NMDA receptor-mediated EPSCs were recorded at holding potential of −70 mV and +50 mV, respectively. Twenty to thirty consecutive traces were averaged to obtain a mean AMPA-EPSC or NMDA-EPSC trace. The amplitude of NMDA-EPSCs was measured 100 ms after the stimulus onset. For measuring I/E ratio, EPSCs and IPSCs were recorded at holding potential of −40 mV and 0 mV, respectively. Twenty to thirty consecutive traces were averaged to obtain a mean EPSC or IPSC trace. mEPSCs were recorded at −70 mV in the presence of 0.1 mM picrotoxin and 0.5 μM TTX. The decay time constant of mEPSC was measured by fitting the decay from the peak to 37% of the peak amplitude with single exponential. mIPSCs were recorded at −70 mV in the presence of 10 μM NBQX, 50 μM D-AP5 and 0.5 μM TTX. To examine the effects of CNTNAP2 or AHI1 knockdown in the mPFC, recordings were made from EGFP-positive (knockdown) and EGFP-negative (control) cells in the same slices.

To evaluate the effects of CX546 on excitatory synaptic transmission, mEPSCs were recorded before application of CX546. CX546 (200 μM) was added to the bathing ACSF for at least 15–20 min, and then mEPSCs were recorded. mIPSCs and I/E ratio were recorded in CX546-untreated and CX546-treated cells separately.

**Immunohistochemistry.** Mice were deeply anesthetized with pentobarbital (100 μg/g of body weight, intraperitoneal injection), perfused with 4% paraformaldehyde (PFA) in 0.1 M phosphate buffer and then processed to obtain coronal sections (100 μm in thickness). After permeabilization (0.5% Triton-X) and blockade of nonspecific binding (10% donkey serum), a rat anti-GFP antibody (1:1000; Nacalai Tesque, Kyoto, Japan), a mouse anti-CaMKII antibody (1:200; Abcam, Cambridge, UK) and a mouse anti-NeuN antibody (1:1000; MerckMillipore, Darmstadt, Germany) were applied overnight at 4 °C. After incubation with secondary antibodies for 4 h at room temperature (an anti-rat IgG Alexa Fluor 488 antibody; 1:400, Life Technologies, Inc. or an anti-mouse Cy5 antibody; 1:300, Jackson), the immunolabeled sections were counterstained with DAPI (300 nM; Invitrogen, Life Technologies, Inc.) for nucleic acid staining.

**Evaluation of knockdown efficacy.** We used the HEK293T cell system[70,75] to evaluate the efficacy of CNTNAP2 or AHI1 knockdown. HEK293T cells in a 24-well dish were transfected with an mOrange2-fused cDNA (CNTNAP2-mOrange or AHI1-mOrange) together with one of the following vector, an RNAi-knockdown (CNTNAP2-knockdown or AHI1-knockdown) vector, a rescue (CNTNAP2-rescue or AHI1-rescue) vector, or a scramble (CNTNAP2-Scr or AHI1-Scr) vector, using X-tremeGENE 9 reagents. One day later, these cells were

fixed. After permeabilization by Triton-X-100 and blocking, primary antibodies (a Rat anti-GFP antibody, Nacalai Tesque; a Rabbit anti-DsRed antibody, Clontech laboratories inc.) were applied. Fluorescence signals from these cells were examined under a confocal laser scanning microscope (FV1200, Olympus).

**In situ hybridization.** Mouse cDNA fragments of CNTNAP2 (nucleotides 2200–3900; GenBank accession number NM_001004357.2), AHI1 (2310–3508; GenBank accession number NM_026203.3), and VGluT1 (301–1680; BC054462) were used. Digoxigenin (DIG)- or fluorescein-labeled cRNA probes were prepared for simultaneous detection of multiple mRNAs by fluorescence in situ hybridization[76]. In brief, fresh frozen sections were hybridized with a mixture of DIG- or fluorescein-labeled cRNA probes. After a stringent posthybridization wash, DIG and fluorescein were detected using a two-step method as follows: the first detection was performed using a peroxidase-conjugated anti-fluorescein antibody (Invitrogen) plus the FITC-TSA plus amplification kit (PerkinElmer, MA, USA); the second detection was performed with a peroxidase-conjugated anti-DIG antibody (Roche Diagnostics, Mannheim, Germany) and the Cy3-TSA plus amplification kit (PerkinElmer).

**In utero electroporation.** For in utero electroporation[44,50], pregnant ICR mice at embryonic day (E) 14–15 were deeply anesthetized with sodium pentobarbital (Somnopentil; Kyoritsu Seiyaku Co., Tokyo, Japan) injected intraperitoneally. Plasmids were dissolved in water. The pCAG plasmid vectors were used to knockdown the expression of CNTNAP2 or AHI1 in pyramidal neurons. The plasmids (0.5–2 μg/μl) were injected into either the left or the right lateral ventricle, or bilateral ventricles with a glass micropipette. Electric pulses (35 V for 50 ms, five to ten times at 950 ms intervals) were delivered through forceps-shaped electrodes (CUY650P5, Unique Medical Imada, Aichi, Japan) connected to an electroporator (CUY21, Nepa Gene, Chiba, Japan). Pups that expressed strong fluorescence in the prefrontal cortex of both hemispheres were used for behavior experiments.

**Quantitative analysis of EGFP-positive cells in the cerebral cortex.** To estimate the spatial distribution of cells transfected by in utero electroporation, an EGFP-expression construct was transfected bilaterally into cells in the ventricular zone of mouse pups at E14-E15. After the mice grew up to 8–18 weeks of age, they were perfused under deep pentobarbital anesthesia, coronal forebrain sections (from Bregma: +2.34 mm, +1.94 mm, +1.18 mm, +0.98 mm, and −1.34 mm) were obtained, and images of the sections were acquired with a confocal laser scanning microscope (FV1200, Olympus). Fluorescence signals in medial prefrontal cortex (mPFC), dorsolateral prefrontal cortex (dl-PFC), orbitofrontal cortex (OFC), motor cortex (mCX), sensory cortex (sCX), cingulate cortex (cgCX), retrosplenial cortex (rsCX), and other cortical areas were calculated by using an image analysis software (ImageJ, NIH). A mouse brain atlas (The Mouse Brain in Stereotaxic Coordinates, second edition) was used to define these cortical areas. The percentage of fluorescence signals of each area relative to the total intensity of all the areas was calculated.

To determine the transfection rate into pyramidal neurons by in utero electroporation, the number of double-labeled cells for EGFP (the transfection marker) and CaMKII (a marker for pyramidal neurons) was counted in the regions of the PFC where EGFP-positive cells were dense in layer 2/3. Then the ratio of the number of double-labeled cells to that of CaMKII-positive cells, the ratio of double-labeled cells to that of EGFP-positive cells, and the ratio of the number of EGFP-positive cells to that of CaMKII-positive cells were calculated.

**Behavioral assays.** All behavioral assays except for the ultrasonic vocalization test were performed with age-matched adult male mice (8–18 weeks). All behavioral studies were performed during light-on periods. Prior to all experiments, mice were acclimatized to the behavioral testing areas for at least 60 min. CNTNAP2-scramble and AHI1-scramble mice were used as control for CNTNAP2-knockdown (CNTNAP2-KD) and AHI1-knockdown (AHI1-KD) mice, respectively. Unless otherwise noted, analyses were performed automatically. To analyze behavioral data in a blind fashion, an independent person numbered randomly all of the related data files (1 to N). N is the number of the related behavioral data files (Sociality test, Three chamber sociality test, Reciprocal social interaction test, Grooming behavior, Novel object test). The experimental conditions of the test mouse (with or without knockdown, treated with vehicle or CX546) were withheld to an observer who manually analyzed the data until after the analyses were complete.

**Sociality test.** An open field apparatus (50 × 40 × 50 cm) was used for the sociality test[77]. In the first session of the test, two empty cylindrical wire cages (10 cm in diameter, 15 cm high) were located in two adjacent corners and a test mouse was placed in the open field for 10 min for the mouse to be habituated to it. In the second session, a novel mouse (adult male C57/B6, SLC JAPAN) was placed in one of the two-wire cages and the other wire cage was kept empty (empty cage). Then, the test mouse was placed in the open field and allowed to freely explore the open field for 10 min. The two sessions were recorded by video. The time spent in the two corner squares (16.7 × 16.7 cm) containing the cages was measured and the time sniffing the novel mouse containing cages were measured manually by an observer who remained blind to the experimental conditions of the test mouse

(with or without knockdown, and treated by vehicle or CX546) throughout analysis. The cages were cleaned with water between each session.

**Three chamber sociality test**. The three-chamber sociality test was adapted from Yang et al.[78–81]. A three-chamber apparatus (50 × 42 cm, equipped with dividing walls with doorway (13 cm width) which enable mice to access each chamber) was used. In the first session of the test, two empty cylindrical wire cages (10 cm in diameter, 15 cm high) were located in both side chambers. A test mouse was placed in the center chamber and allowed to explore the two-wire cages for 10 min. In the second session, a novel mouse (adult male C57/B6, SLC JAPAN) was placed in one of the two-wire cages and a novel object with similar color and similar volume to the novel mouse was newly placed in the other wire cage. Then, the test mouse was placed in the center chamber and allowed to freely explore the three chambers for 10 min. The two sessions were recorded by video. The time for the test mouse to be facing and sniffing the novel mouse containing cage (T-mouse) and that for the test mouse to be facing and sniffing the novel object containing cage (T-object) were measured by an observer who remained blind to the experimental conditions of the test mouse (with or without knockdown) throughout analysis. The sociality index[78–80] was calculated as [T-mouse − T-object]/ [T-mouse + T-object]. The cages were cleaned with water between each session.

**Reciprocal social interaction test**. For the reciprocal social interaction test[10], mice were placed in the corner of an open field (50 × 40 × 50 cm, 15 lux) and adult male DBA2 mice (SLC JAPAN) were placed in the opposite corner. Behavior was recorded using a video camera located above the open field for 5 min. Time spent for physical contact was analyzed automatically with TimeOF4 software (O'Hara & Co., Japan). The number and the time of active contacts (i.e., the test mouse contacted the DBA2 mouse), the number of following, that of nose-to-anogenital sniffing, that of nose-to-nose sniffing, and that of physical contacts (including both active and passive contacts) were analyzed manually by an observer who remained blind to the experimental conditions of the test mouse (with or without knockdown, and treated by vehicle or CX546) throughout analysis.

**Ultrasonic vocalization**. The Ultrasonic vocalization (USV) test[18] was performed in mice at P7 because the number, duration, and average peak frequency of USVs had a peak around P7[82]. Both male and female pups at P7 were isolated from their mothers at random and gently placed into a container (plastic cylinder, 5 cm high, 5 cm in diameter). USV was recorded for 5 min from the time of isolation. USV emission was monitored by an ultrasonic microphone (UltraSoundGate CM 16; Avisoft Bioacoustics, Berlin, Germany). The isolation container and microphone were placed in a sound attenuating box (63.5 × 42 × 37 cm). The microphone (sensitive to frequencies of 15–250 kHz) was connected to a personal computer with a sampling rate of 250 kHz in 16-bit format by Avisoft RECORDER (version 4.2.16; Avisoft Bioacoustics) via a USB recording interface (UltraSoundGate 116Hb; Avisoft Bioacoustics, Berlin, Germany).

**Grooming behavior**. Mice were placed in a transparent plastic cylinder (22 cm high, 12 cm in diameter). After habituation for 5 min, grooming behavior was video-recorded for 10 min. Grooming involves strokes around nose, face, ears, and head with the forepaws, scratching body with the limbs and licking the body. Grooming time was measured by an observer who remained blind to the experimental conditions of the test mouse (with or without knockdown) throughout analysis.

**Marble bury test**. For the marble bury test[41], twelve clear blue glass marbles (17 mm diameter), white paper bedding material (Shepherd Specialty Papers, Kalamazoo, MI) and transparent cages (20 × 11 × 12.5 cm) were used. The cage was filled with bedding (5 cm depth) and 12 marbles were placed on the bedding surface spaced at ~4 cm intervals. The mice were placed into the cages and allowed to remain for 15 min. The number of marbles buried more than two-thirds in the bedding was counted.

**Operant reversal learning**. Operant chambers which consisted of two retractable levers and a pellet feeder, located in a sound attenuating box, were used for this test (Med-Associates Georgia, VT, USA). Two retractable levers were located symmetrically from the pellet feeder (i.e. Left and right lever). This test consisted of two parts: the training phase (day1–day5) and the reversal phase (day6–day12). In the training phase, mice were trained to press the active lever to obtain food pellets. Mice were fed with one food pellet if they pressed the active, but not inactive, lever. Position of active or inactive lever was assigned randomly for each mouse. Maximum of 20 food pellets were allowed during a 20 min session. Four sessions were repeated in one experimental day. In the reversal phase, the positions of active and inactive levers were swapped from those in the training phase. Four sessions were repeated in one experimental day. The percentage of correct lever press relative to the total lever press was calculated. Through the experiment, mice were restricted in food intake to maintain their body weight 80–90% of that during free-feeding. Restriction of food intake began two days prior to the training. Mice were weighed

after daily sessions and fed an adjusted amount of food to maintain the target body weight. Water was available ad libitum throughout the experiment.

**Novel object test**. An open field apparatus (50 × 40 × 50 cm) was used for the novel object test[77]. In the first session of the test, mice were allowed to explore the open field freely for 10 min as a habituation session. In the second session, a novel cylindrical object (20 cm height, 5 cm diameter, black colored) was located in the center of the open field. Mice were allowed to explore the novel object for 10 min. Time exploring the novel object was measured by an observer who remained blind to the experimental conditions of the test mouse (with or without knockdown) throughout analysis.

**Elevated plus maze test**. The Elevated plus maze (EPM) consisted of closed arms (25 × 5 × 15 cm (H)), open arms (25 × 5 × 0.5 cm (H)) and central platform of the maze (5 × 5 cm)[77]. It was elevated 50 cm above the floor and placed in the light (200 lx). In the EPM test[77], mice were individually placed in the center area and allowed to move freely in the space for 10 min. The percent time spent in the open arms, the number of entry to the open arm and the total distance traveled were measured[77]. Data were collected and analyzed with TimeEP1 software (O'Hara & Co., Japan).

**Light/Dark box test**. For the Light/Dark (LD) box test[77], the apparatus consisted of light (250 lx) and dark chamber (20 × 20 × 25 (H) cm). There was an entrance which enabled mice to move across the light and dark chambers (3 × 5 cm). In the LD test, mice were individually introduced into the dark box and allowed to move freely in the LD box for 10 min. The time spent in the light chamber and the number of transitions between the light and dark chambers were measured. Data were collected and analyzed with TimeLD4 (O'Hara & Co., Japan).

**Open field test**. For the open field test[77], mice were placed in an open field apparatus (50 × 40 × 50 cm) with 15 lx. Mice were recorded with a video camera for 30 min, and the total distance traveled was calculated in 5 min time bins with TimeOF4 software (O'Hara & Co., Japan).

**Tail suspension test**. For the tail suspension test[83], individual mice were suspended by their tails using self-adhesive tapes from a small hook at one end of a perpendicular wire for 10 min. Mouse behavior was recorded with a video camera. The duration of immobility was analyzed automatically with TimeFZ2 software (O'Hara & Co., Japan).

**Forced swim test**. For the forced swim test[83], mice were placed individually in a transparent plastic cylinder (22 cm high, 12 cm in diameter), containing water at 25 ± 0.5 °C at a depth of 15 cm, and were forced to swim for 10 min. Mice behavior was recorded with a video camera. In the latter half (for 5 min) of the 10-min session, duration of immobility was measured automatically with TimeFZ2 software (O'Hara & Co., Japan).

**Y-maze test**. For the Y-maze test[84], mice were placed at the end of one arm with the brightness of 200 lx and allowed to move freely throughout the maze for 7 min. The series of arm entries were analyzed automatically with TimeYM1 software (O'Hara & Co., Japan). If a mouse consecutively enters each of the three arms with no repetitions, it was defined as alternation. The percent alternation was calculated as [number of alternations]/[(total number of arm entries − 2)×100] (%).

**Pharmacological rescue**. CX546 (Sigma, USA) was dissolved in 25% β-cyclodextrin (Nacalai Tesque, Kyoto, Japan) in water[20]. The dose of CX546 was determined based on the previous reports[20,85]. We employed acute CX546 treatment because previous studies showed that impaired social interaction was improved by acute injection of AMPA receptor agonist or positive allosteric modulator in ASD mouse models[19,20]. Intraperitoneal injection of CX546 (25 mg per kg) or vehicle (25% β-cyclodextrin in water) was performed 30 min before the reciprocal social interaction test or 20 min before the sociality test, because there was a 10 min habituation period in the latter test.

**Statistics**. Data were expressed as the mean ± SEM. The Mann–Whitney $U$ test was used to analyze the electrophysiological data when two independent data sets were compared in pharmacological experiments. For multiple comparisons, the Dunn test was used to analyze the electrophysiological and knockdown efficacy data. Unpaired Student's $t$ test and paired $t$ test were used to determine statistical significance for the behavioral data. Differences between groups were judged to be significant when $P < 0.05$. All statistical analyses were performed using Graphpad Prism software (Graphpad software, San Diego, CA, USA).

**Reporting summary**. Further information on research design is available in the Nature Research Reporting Summary linked to this article.

## Data availability

The authors declare that the data supporting the findings of this study are available within the article, its Supplementary Information files, the Source Data file or from the corresponding authors upon reasonable request.

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

## Acknowledgements

We thank K. Kitamura, T. Nakazawa, Y Hashimotodani, and Y. Sugaya for helpful discussion, and K. Matsuyama, M. Sekiguchi, M. Watanabe and M. Baba for technical assistance. This work was supported by Grants-in-Aid for Scientific Research (25000015, 18H04012, 19H05204 to M.K., 24220007 to M.W., and 18H02539 to N.U.) from JSPS, Japan, by Takeda Science Foundation (N.U.), and by AMED under Grant Number JP20dm0107091.

## Author contributions

H.Sacai., N.U., and M.K. designed the study. H.Sacai., K.S., K.K., H.Suzuki., K.N., T.W., M.W., N.U., and M.K. performed the experiments and/or data analysis. H.Sacai., N.U., and M.K. wrote the paper.

## Competing interests

The authors declare no competing interests.

## Additional information

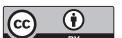

