## [Peer Review File · Nature Communications]

Reviewers' comments:

Reviewer #1 (Remarks to the Author):

The authors use knockdown of 2 autism risk genes to study the role of synaptic physiology in L2/3 neurons of the PFC and how this relates to social behavior. They claim that both models have reduced excitatory neurotransmission, and that pharmacological enhancement of AMPAR function restored impaired social behavior in both models. The data support both claims.

This is a well-done study. There are many strengths. They compare two mouse models with different mechanisms of action. The use of a scrambled RNAi is a good control for the RNAi group. The use of an RNAi-resistant constructs to rescue the knockdown is also a nice control. That they measured electrophysiological phenotypes in adolescent (2-3 week old) and adult (8 -11 week old) is also very nice. Their behavioral experiments are well-powered for the differences they are detecting between groups. They have physiology and behavior in both models and perform a rescue experiment to test a hypothesis about synaptic transmission generated by the literature and their first round of studies.

This will be of interest to people in the field of autism and neurodevelopmental disorders in general, as it adds to our understanding of how genetic changes disrupt neurophysiology to influence behavior.

I think that with inclusion of some more detail, the report will be quite solid.

Most importantly, the authors do not show convincing evidence that their L2/3 selective targeting was indeed selective for L2/3. In Figure 1, I can't tell where L2/3 begins and ends, and there is no quantification of the representative images. The specificity for L2/3 was really hard to discern with the low magnification images showing in Fig S2 and then the super high magnification in Fig S3. It would be helpful to show an in-between magnification image so the reader can assess for themselves whether the staining is in L2/3. We also want to know how many cells were transfected in Layers 1, 5, and 6.

"N=10 images" is not an appropriate sampling unit for statistics. How many slices? How many animals?

In addition to the quantification in Fig S3b "GFP + CaMKII positives cells / CaMKII positive cells (%)", I'd want to know 1) what proportion of GFP+ cells are also CaMKII+, and 2) what proportion of CaMKII+ cells are also GFP+?

Finally, do you see uptake and staining in L2/3 throughout cortex? What about other brain regions? This is important because your AMPA modulation experiment is a systemic treatment, and therefore will change glutamatergic signaling in brain regions other than mPFC. So, we can't really conclude that your systemic administration of CX546 had its behavioral effect based on changes to glutamatergic signaling specifically in PFC.

Physiology convincingly suggests that the number of synapses onto mPFC L2/3 neurons is decreased in Cntnap2 KD cells (based on decreased mini frequency, unchanged mini amplitude, no change in PPR), and that this persists into adulthood (8-11 weeks old). They see a similar change in inhibitory synapses (decreased mini frequency without change in amplitude; PPR not assessed). AMPA/NMDA ratio unchanged.

For the AHI1 gene, they convincingly show that knockdown in L2/3 PFC neurons causes increased NMDA/AMPA ratio, decreased excitatory mini amplitude; unchanged excitatory mini frequency. The HAP1 experiments are also nice to give a plausible mechanism for AHI1 function.

For mPFC recordings, were you recording from ACC, PL, and/or IL?

For behavior experiments, I strongly recommend showing individual animal data points in your summary figures. I also recommend providing a table of your means +/- SEM & p values because I don't see this reported anywhere. It will help the reader be convinced that there really is no anxiety phenotype, because by eye it looks like the Cntnap2 KD mice spend less time in the open arm than controls do.

Overall, the statistical treatments seem appropriate, and the work is reproducible given the level of detail provided.

Points that could make the discussion more robust:

Do you think the impaired inhibitory synaptic transmission is causal or compensatory? How do your data relate to measures of intrinsic neuronal excitability in pyramidal neurons in ASD models in PFC, and recent E/I balance work by Dan Feldman's group in somatosensory cortex?

Minor suggestions for improvement:

Introduction: "We performed knockdown of Cntnap2 or Ahi1 in layer II/III pyramidal neurons of the mouse prefrontal cortex (PFC), which has been implicated in social behavior and ASD^{21,27-30}" This implies that the references cited support the role of L2/3 in ASD; these references only support the role of the PFC. Please provide additional references here to support the rationale for why you are studying L2/3 cells as opposed to other cell populations in this brain regions.

I don't think that Yizhar et al. 2011 is the best example of your statement "it has been thought that synaptic dysfunction underlies the pathophysiology of ASD (21). Consider replacing with another paper that really tests synaptic dysfunction.

It was really difficult to understand the sentence in the Fig S1 legend that goes on for 11 lines. I was unable to understand what went with what and what the goal of each manipulation was. Please consider breaking into shorter sentences and state the goal of each manipulation.

Why was mOrange not visualized or quantified?

Did you test USV's for both models or just the one?

P33,L4 - "open field test" misspelling

P33,L5 - "Mice were placed in an open field apparatus (50 x 40 x 50 cm) with 15 lx." What is 15 lx?

Why do reference skip from 1,2 to 9-15?

Audrey C. Brumback, MD, PhD
Dell Medical School at UT Austin

Reviewer #2 (Remarks to the Author):

This interesting manuscript addresses the hypothesis that knockdown of Cntnap2 or Ahi1 during postnatal days 16 to 24 reduces excitatory neurotransmission in the prefrontal cortex and produces autism-relevant behaviors in mice. CNTNAP2 mutations have been identified in a few people with autism spectrum disorder. AHI1 mutations are stated as causally implicated in Joubert syndrome, in which autism is associated in about 30% of cases. Studies focus on layer II/III pyramidal neurons, and test the effects of treatment with the AMPA receptor agonist CX546 on restoring electrophysiological and behavioral abnormalities in the two mouse models.

The rationale for the electrophysiological experiments is well described. Appropriate controls for the knockdowns were used, *Cntnap2*-scramble and *Ahi1*-scramble mice. The authors are to be complimented for conducting a large number of behavioral assays to investigate phenotypes relevant to the diagnostic and associated symptoms of autism.

Unfortunately, serious conceptual flaws in the methods make the key behavioral results uninterpretable.

Major issues

1. Social approach test

a) This test is supposed to measure time spent with a novel object versus time spent with a novel mouse. Because both wire cages were present during habituation in the present experiments, the subsequent test session compared time spent around a familiar object versus time spent around a novel mouse. Most mice prefer novelty. As conducted, the assay measured novelty detection, not sociability.

b) "Stay time" is not an informative parameter. The tracking system must be able to determine whether the subject mouse was facing the mouse cage and facing the empty cage. The subject mouse could be in close proximity, but facing away, therefore not exploring the targets.

c) Statistical comparisons are incorrectly conducted. The meaningful comparison is time spent with the novel mouse versus time spent with the novel object, WITHIN genotype. From the graphs shown in Figures 2, 4, and 6, both control and *Cntnap2* mice spent more time with the novel mouse than with the familiar object. Both control and *Ahi1* mice spent more time with the novel mouse than with the familiar object. The knockdown therefore did not produce a social deficit. There is no social abnormality signal to use in evaluating drug reversal.

2. Reciprocal social interactions

a) "Number of active contacts" is insufficiently explained. Freely moving mouse dyads engage in following, nose-to-anogenital sniffing, nose-to-nose sniffing, and physical contact. Each parameter should be separately scored and displayed in separate graphs.

b) No information is provided about scoring methods by the human observers.

c) No information is provided about the specific methods used to ensure that the human observers remained blind to genotype and treatment condition while scoring.

d) Although a significant difference was detected in number of ultrasonic vocalizations between genotypes, the magnitude of the differences was small, and unlikely to be biologically important.

e) Ultrasonic vocalizations emitted by separated pups are analogous to babies crying, which elicit responses by the parents and thereby meet the definition of communication. In autism, communication deficits are primarily in speech and interactive conversation during childhood and adulthood, not abnormal infant crying. Although the pup vocalization assay has been frequently used by molecular genetics labs working with mouse models of autism, it does not provide a measure of communication with high relevance to autism.

Ultrasonic vocalizations in juvenile and adult mice during social interactions offer better measures for mouse models of autism. However, the communication function of USVs in juvenile and adult mice has not been definitively established. Mice appear to communicate primarily through social odors.

The authors are encouraged to read the original literature on mouse social and communication tasks relevant to autism, and adjust their methods for future studies.

3. Self-grooming

As above, no information is provided about scoring methods, definitions of grooming, and methods to ensure blind rating.

4. Light/dark box

The two parameters relevant to anxiety-related behaviors in this assay are (a) time spent in the light chamber and (2) number of transitions between the light and dark chambers. Latency to the first move into the light box, and the total distance traveled, can be removed from the data presentations.

5. Forced swim

Usually the first 5 minutes in the water are excluded from scoring.

6. Y-maze

Page 34 methods state "If a mouse consecutively enters one of the three arms with no repetitions, it was defined as alternation." This seems incorrect. Did the authors mean to say "If a mouse consecutively enters EACH of the three arms with no repetitions it was defined as alternation."?

7. AMPA drug treatment

Methods page 34 states that CX546 was administered "at a dose of 25 mg per kg or vehicle (25% b-cyclodextrin in water) 30 min before the reciprocal social interaction test or 20 min before the social novelty test."

(a) Please justify the choice of the dose used. Describe and cite the literature on behavioral effects of CX546 in mice.

(b) Please justify the choice of acute treatment. Describe and cite the literature on the effects of acute versus chronic CX546 treatment on behaviors in mice.

Minor issues

1. English grammar and spelling requires corrections throughout the text. As examples:

(a) Second sentence of Abstract: "Many genes with mutation in ASD patients have been identified, ..."

(b) Page 34 "Statistic," "Student's t-test was used to the behavioral data."

2. The end of the Introduction repeats the findings stated in the Abstract and Discussion. A better use of the Introduction would be to describe the literature on CNTNAP2 mutations in people with autism, the symptoms of people with AHI1 mutations, and the existing literature on mouse models.

(a) More detailed information on the behavioral phenotypes of Cntnap2 and Ahi1 mice would be useful. There has been considerable controversy around failures to replicate the social deficits originally reported (reference 3, Peñagarikano et al.) in Cntnap2 mice, probably due to using lines from different sources and different generations of backcrosses.

(b) Literature from other mouse models of autism on excitatory and behavioral abnormalities, and on Ampakine treatments, would be useful to describe and cite in the Introduction.

(c) The human CNTNAP2 literature primarily describes polymorphisms. The original studies cited were restricted to an Amish population in Pennsylvania, USA. Later studies implicated CNTNAP2 in other disorders including speech and intellectual disabilities.

3. Figure 1: Please remove the diagrams shown in panels b, e, g. These are standard tasks, which do not require illustrations. The diagrams are charming but unnecessary. Panel a could remain, since the authors used a non-standard apparatus to measure social approach.

Reviewer #3 (Remarks to the Author):

This manuscript reports that reduced excitatory synaptic transmission in layer II/III pyramidal neurons in the prefrontal cortex (PFC) in two independent mouse models of ASD is responsible for the reduced social interaction. In support of these conclusions, the authors demonstrate that the I/E ratio in layer II/III pyramidal neurons in mice knocked down of *Cntnap2* and *Ahi1* by in utero electroporation is increased. This synaptic phenotype is associated with suppressed social interaction, which is rescued by CX546 (ampakine) that promotes excitatory synaptic transmission in the two independent mice.

Given that E/I imbalance has been implicated in ASD, and mPFC in the brain, a brain region associated with higher brain functions, has not been extensively studied, these results provide significant insights into the how altered E/I balance in the mPFC could affect social behavior in mouse models of ASD.

Major comments:

1. Fig. 5 evaluates the impacts of CX546 on the decay kinetics and charge transfer of mEPSCs but does not test whether the E/I ratio, likely the most important parameter, is normalized.

2. In addition, does CX546 have any effects on mIPSCs in layer II/III pyramidal neurons? GABAergic neurons have excitatory synapses on their dendrites, and thus it is likely that CX546 also affects the E/I balance and output functions in these neurons.

Minor comments:

1. It is unclear why the authors focused on layers II/III but not other cortical layers such as layer V.

2. In Fig. 1a, it is unclear whether *Cntnap2* is expressed only layer II/III or other layers.

3. It is unclear how the electroporation specifically targets layer II/III pyramidal neurons but not other cortical layers. This should be clarified.

4. Fig. 1b,c. The social novelty test referred by the authors seem to be social interaction or sociality test because this test evaluates only whether the subject mouse prefers social or empty target (not old or new social target).

5. Fig. 2e,f and Fig. 4d. Total duration of social interaction in addition to contacting number should also be shown.

6. Fig. 2b,c. Is there any reason why the authors did not use the well-known three-chamber test? In addition, the way the data in Fig. 2d could also be presented in a way by comparing the stay

time of M and E because M and E are competing in the same session. The same applies to other figures that show similar results.

7. Fig. 2g,h: Ideally, several early postnatal stages in addition to just P7 should be used to evaluate ultrasonic vocalizations.

8. CX546, which increases excitatory synaptic transmission in WT slices, does not affect social behavior in WT mice (Figs 5 and 6). This should be explained.

9. The authors need to cite previous papers that investigated the roles of PFC neurons and their synaptic transmission on social behavior in ASD models (i.e. PMID: 30659288; PMID: 28768803; PMID: 25824299 etc).

10. The following conclusion on page 8 is too strong: "The results so far indicate that Cntnap2 knockdown in layer II/III pyramidal neurons of the developing mouse PFC reduces both excitatory and inhibitory 1 synaptic inputs to these neurons, increases I/E balance, and causes ASD-like behavior."

Points of Revision

Responses to reviewers:

We thank the reviewers for critically reading our manuscript and raising several important issues. We tried to address all the comments by the reviewers and followed their suggestions as far as possible. We feel that the manuscript has been improved significantly. We have indicated the changes in the main text, Figure and Supplemental Figure with red so that they are easily identifiable.

Responses to Reviewer #1:

(Comments)

The authors use knockdown of 2 autism risk genes to study the role of synaptic physiology in L2/3 neurons of the PFC and how this relates to social behavior. They claim that both models have reduced excitatory neurotransmission, and that pharmacological enhancement of AMPAR function restored impaired social behavior in both models. The data support both claims.

This is a well-done study. There are many strengths. They compare two mouse models with different mechanisms of action. The use of a scrambled RNAi is a good control for the RNAi group. The use of an RNAi-resistant constructs to rescue the knockdown is also a nice control. That they measured electrophysiological phenotypes in adolescent (2-3 week old) and adult (8 -11 week old) is also very nice. Their behavioral experiments are well-powered for the differences they are detecting between groups. They have physiology and behavior in both models and perform a rescue experiment to test a hypothesis about synaptic transmission generated by the literature and their first round of studies.

This will be of interest to people in the field of autism and neurodevelopmental disorders in general, as it adds to our understanding of how genetic changes disrupt neurophysiology to influence behavior.

I think that with inclusion of some more detail, the report will be quite solid.

(Our response)

We appreciate Reviewer #1's constructive suggestions and comments. Our point-by-point responses to the comments are as follows.

(Comments)

Most importantly, the authors do not show convincing evidence that their L2/3 selective targeting was indeed selective for L2/3.... It would be helpful to show an in-between magnification image so the reader can assess for themselves whether the staining is in L2/3. We also want to know how many cells were transfected in Layers 1, 5, and 6.

(Our response)

We appreciate the reviewer's helpful suggestions. Previous studies demonstrate that genes are selectively expressed in layer II/III pyramidal neurons after in utero electroporation at embryonic day 14-15 (Niwa et al., *Neuron* **65**, 480-489, 2012, Bitzenhofer et al., *Front Cell Neurosci* **11**, 239, 2017, Bitzenhofer et al., *Nat Commun* **8**, 14563, 2017). We have added in-between magnification images and show that GFP-positive transfected cells were confined to layer II/II of the PFC and there were few GFP-positive cells in layer I, V and VI. We now present and explain these data in Supplementary Fig. 4 and in the Results section (page 6 line 21-22).

- Niwa M1, Kamiya A, Murai R, Kubo K, Gruber AJ, Tomita K, Lu L, Tomisato S, Jaaro-Peled H, Seshadri S, Hiyama H, Huang B, Kohda K, Noda Y, O'Donnell P, Nakajima K, Sawa A, Nabeshima T. *Neuron*. 2010 Feb 25;65(4):480-9. doi: 10.1016/j.neuron.2010.01.019. Knockdown of DISC1 by in utero gene transfer disturbs postnatal dopaminergic maturation in the frontal cortex and leads to adult behavioral deficits.
- Bitzenhofer SH1, Ahlbeck J1, Hanganu-Opatz IL1. *Front Cell Neurosci*. 2017 Aug 14;11:239. doi: 10.3389/fncel.2017.00239. eCollection 2017. Methodological Approach for Optogenetic Manipulation of Neonatal Neuronal Networks.
- Bitzenhofer SH1, Ahlbeck J1, Wolff A1, Wiegert JS2, Gee CE2, Oertner TG2, Hanganu-Opatz IL1. *Nat Commun*. 2017 Feb 20;8:14563. doi: 10.1038/ncomms14563. Layer-specific optogenetic activation of pyramidal neurons causes beta-gamma entrainment of neonatal networks.

(Comments)

“N=10 images” is not an appropriate sampling unit for statistics. How many slices? How many animals?

(Our response)

We appreciate the reviewer’s helpful suggestion. We examined 5 slices from 3 mice. We have carefully revised sampling units throughout the text and newly made a summary table for describing all the data.

(Comments)

In addition to the quantification in Fig S3b “GFP + CaMKII positives cells / CaMKII positive cells (%)”, I’d want to know 1) what proportion of GFP+ cells are also CaMKII+, and 2) what proportion of CaMKII+ cells are also GFP+?

(Our response)

Following the reviewer’s suggestion, we have quantified the following two values, “GFP and CaMKII double positive cells / GFP positive cells” and “GFP positive cells / CaMKII positive cells”. We found that the percentage value of “GFP and CaMKII double positive cells / GFP positive cells” was approximately 100% whereas that of “GFP positive cells / CaMKII positive cells” was around 25%. We present and explain these data in Supplementary Fig. 5c and d and in the Results section (page 6 line 24-page 7 line 3).

(Comments)

Finally, do you see uptake and staining in L2/3 throughout cortex? What about other brain regions? This is important because your AMPA modulation experiment is a systemic treatment, and therefore will change glutamatergic signaling in brain regions other than mPFC. So, we can’t really conclude that your systemic administration of CX546 had its behavioral effect based on changes to glutamatergic signaling specifically in PFC.

(Our response)

We thank the reviewer for raising an important issue. We cannot rule out the possibility that social deficits might be improved by enhancement of glutamatergic signaling in brain regions other than the mPFC. However, mice transfected with scramble miRNA displayed normal sociality after the systemic administration of CX546 (Fig. 6). In contrast, knockdown mice commonly had reduced excitatory transmission in layer II/III pyramidal neurons of the PFC (Fig. 1c-g and 3b-h) and restored the impaired sociality by the systemic administration of CX546 (Fig. 6). Moreover, transfected cells in which excitatory transmission was reduced were mainly present in layer II/III of the PFC (Supplementary Fig. 3, 4 and 11). Therefore, we assume that the effects of CX546 on social behavior are based mainly on enhanced excitatory transmission in layer II/III pyramidal neurons of the PFC. We now present an explanation for the effects of CX546 in the Results section (page 13 line 17-22 and page 14 line 1-7).

(Comments)

Physiology convincingly suggests that the number of synapses onto mPFC L2/3 neurons is decreased in Cntnap2 KD cells (based on decreased mini frequency, unchanged mini amplitude, no change in PPR), and that this persists into adulthood (8-11 weeks old). They see a similar change in inhibitory synapses (decreased mini frequency without change in amplitude; PPR not assessed). AMPA/NMDA ratio unchanged.

For the AH11 gene, they convincingly show that knockdown in L2/3 PFC neurons causes increased NMDA/AMPA ratio, decreased excitatory mini amplitude; unchanged excitatory mini frequency. The HAPI experiments are also nice to give a plausible mechanism for AH11 function.

For mPFC recordings, were you recording from ACC, PL, and/or IL?

(Our response)

Yes. We recorded pyramidal neurons in the ACC, PL, and/or IL. We present additional information in the Methods section (page 30 line 20).

(Comments)

For behavior experiments, I strongly recommend showing individual animal data points in your summary figures. I also recommend providing a table of your means +/- SEM & p values because I don't see this reported anywhere. It will help the reader be convinced that there really is no anxiety phenotype, because by eye it looks like the Cntnap2 KD mice spend less time in the open arm than controls do.

(Our response)

We have followed the reviewer's advice and now present individual data points and their average in all graphs. Furthermore, we provide a summary table of means +/- SEM, p values and statistics.

(Comments)

Overall, the statistical treatments seem appropriate, and the work is reproducible given the level of detail provided.

Points that could make the discussion more robust:

Do you think the impaired inhibitory synaptic transmission is causal or compensatory? How do your data relate to measures of intrinsic neuronal excitability in pyramidal neurons in ASD models in PFC, and recent E/I balance work by Dan Feldman's group in somatosensory cortex?

(Our response)

We appreciate this valuable comment. We think that the impaired inhibitory synaptic transmission is not causal for ASD-like behaviors. CNTNAP2 knockdown reduced both excitatory and inhibitory synaptic transmission (Fig. 1c-l), whereas AHI1 knockdown reduced only excitatory transmission (Fig. 3b-h). If the impaired inhibitory transmission in CNTNAP2 knockdown mice is causal for ASD-like behaviors, inhibitory transmission might also be impaired by AHI1 knockdown. It also supports no causal relationship that CX546 rescued ASD-like abnormal behaviors in CNTNAP2 knockdown mice despite the fact that CX546 did not normalize inhibitory transmission in CNTNAP2 knockdown mice (Fig. 5f). In addition, we assume that the impaired inhibition in CNTNAP2

knockdown mice could not be compensatory for the decreased excitation because AHI1 knockdown mice did not show impaired inhibition despite the decreased excitation. We describe these points with the relationship between our results and intrinsic neuronal excitability as well as the study by Dan Feldman's group in the Results (page 13 line 3-11) and the Discussion (page 16 line 1-6, page 16, line 10-18 and page 18 line 21-page 22 line 4).

(Comments)

Minor suggestions for improvement:

Introduction: "We performed knockdown of Cntnap2 or Ahi1 in layer II/III pyramidal neurons of the mouse prefrontal cortex (PFC), which has been implicated in social behavior and ASD21,27-30"

This implies that the references cited support the role of L2/3 in ASD; these references only support the role of the PFC. Please provide additional references here to support the rationale for why you are studying L2/3 cells as opposed to other cell populations in this brain regions.

(Our response)

We appreciate the reviewer's helpful suggestions. Layer II/III was chosen because (1): CNTNAP2 knockout mice and other mouse models of ASD displayed synapse dysfunction and abnormal network activity in layer II/III pyramidal neurons of the cortex, and (2): ASD patients had abnormal spine morphology in layer II of the prefrontal cortex (Brodmann area 9). We have cited following studies of layer II/III cells in ASD in the Introduction section (page 4 line 14).

- Amodio, D.M., and Frith, C.D. (2006). Meeting of minds: the medial frontal cortex and social cognition. *Nat Rev Neurosci* 7, 268-277.
- Hutsler, J.J., and Zhang, H. (2010). Increased dendritic spine densities on cortical projection neurons in autism spectrum disorders. *Brain Res* 1309, 83-94.
- Tada, H., Miyazaki, T., Takemoto, K., Takase, K., Jitsuki, S., Nakajima, W., Koide, M., Yamamoto, N., Komiya, K., Suyama, K., et al. (2016). Neonatal isolation augments social dominance by altering actin dynamics in the medial prefrontal cortex. *Proc Natl Acad Sci U S A* 113, E7097-E7105.

- Selimbeyoglu, A., Kim, C.K., Inoue, M., Lee, S.Y., Hong, A.S.O., Kauvar, I., Ramakrishnan, C., Fenno, L.E., Davidson, T.J., Wright, M., et al. (2017). Modulation of prefrontal cortex excitation/inhibition balance rescues social behavior in CNTNAP2-deficient mice. *Sci Transl Med* 9.
- Courchesne, E., Mouton, P.R., Calhoun, M.E., Semendeferi, K., Ahrens-Barbeau, C., Hallet, M.J., Barnes, C.C., and Pierce, K. (2011). Neuron number and size in prefrontal cortex of children with autism. *JAMA* 306, 2001-2010.
- Liang, J., Xu, W., Hsu, Y.T., Yee, A.X., Chen, L., and Sudhof, T.C. (2015). Conditional neuroligin-2 knockout in adult medial prefrontal cortex links chronic changes in synaptic inhibition to cognitive impairments. *Mol Psychiatry* 20, 850-859.
- Penagarikano, O., Abrahams, B.S., Herman, E.I., Winden, K.D., Gdalyahu, A., Dong, H., Sonnenblick, L.I., Gruver, R., Almajano, J., Bragin, A., et al. (2011). Absence of CNTNAP2 leads to epilepsy, neuronal migration abnormalities, and core autism-related deficits. *Cell* 147, 235-246.
- Santini, E., Huynh, T.N., MacAskill, A.F., Carter, A.G., Pierre, P., Ruggero, D., Kaphzan, H., and Klann, E. (2013). Exaggerated translation causes synaptic and behavioural aberrations associated with autism. *Nature* 493, 411-415.
- Scearce-Levie, K., Roberson, E.D., Gerstein, H., Cholfin, J.A., Mandiyan, V.S., Shah, N.M., Rubenstein, J.L., and Mucke, L. (2008). Abnormal social behaviors in mice lacking Fgf17. *Genes Brain Behav* 7, 344-354.
- Yizhar, O., Fenno, L.E., Prigge, M., Schneider, F., Davidson, T.J., O'Shea, D.J., Sohal, V.S., Goshen, I., Finkelstein, J., Paz, J.T., et al. (2011). Neocortical excitation/inhibition balance in information processing and social dysfunction. *Nature* 477, 171-178.

(Comments)

I don't think that Yizhar et al. 2011 is the best example of your statement "it has been thought that synaptic dysfunction underlies the pathophysiology of ASD (21). Consider replacing with another paper that really tests synaptic dysfunction.

(Our response)

We thank the reviewer for this suggestion. We replace *Yizhar et al. 2011* with the following papers related to synaptic dysfunction underlying ASD-like behavior (page 3 line 10).

- Chao, H.T., Chen, H., Samaco, R.C., Xue, M., Chahrour, M., Yoo, J., Neul, J.L., Gong, S., Lu, H.C., Heintz, N., et al. (2010). Dysfunction in GABA signalling mediates autism-like stereotypies and Rett syndrome phenotypes. *Nature* 468, 263-269.

- Delorme, R., Ey, E., Toro, R., Leboyer, M., Gillberg, C., and Bourgeron, T. (2013). Progress toward treatments for synaptic defects in autism. *Nat Med* 19, 685-694.
- Peca, J., Feliciano, C., Ting, J.T., Wang, W., Wells, M.F., Venkatraman, T.N., Lascola, C.D., Fu, Z., and Feng, G. (2011). Shank3 mutant mice display autistic-like behaviours and striatal dysfunction. *Nature* 472, 437-442.
- Tabuchi, K., Blundell, J., Etherton, M.R., Hammer, R.E., Liu, X., Powell, C.M., and Sudhof, T.C. (2007). A neuroligin-3 mutation implicated in autism increases inhibitory synaptic transmission in mice. *Science* 318, 71-76.
- Won, H., Lee, H.R., Gee, H.Y., Mah, W., Kim, J.I., Lee, J., Ha, S., Chung, C., Jung, E.S., Cho, Y.S., et al. (2012). Autistic-like social behaviour in Shank2-mutant mice improved by restoring NMDA receptor function. *Nature* 486, 261-265.

(Comments)

It was really difficult to understand the sentence in the Fig S1 legend that goes on for 11 lines. I was unable to understand what went with what and what the goal of each manipulation was. Please consider breaking into shorter sentences and state the goal of each manipulation.

(Our response)

We apologize for our unclear description about the former Supplementary Figure 1. We have rewritten the legend with shorter sentences and by adding the goal of each manipulation in new Supplementary Figure 2.

(Comments)

Why was mOrange not visualized or quantified?

(Our response)

mOrange was visualized and quantified in supplementary Fig. 2 (previous supplementary Fig. 1). We used mOrange-fused cDNA and mOrange-fused RNAi-resistant cDNA. For example, Control in Supplementary Fig. 2a means mOrange-fused CNTNAP2 (top) with a GFP expression vector (bottom) (the 1st columns from the left), CNTNAP KD means mOrange-fused CNTNAP2 (top) and CNTNAP2 KD vector with GFP (bottom), CNTNAP2 Res means mOrange-fused RNAi-resistant CNTNAP2 (top) and CNTNAP2 KD vector with GFP (bottom), and CNTNAP2 Scr means mOrange-fused CNTNAP2 (top) and CNTNAP2 Scr vector with GFP. If CNTNAP2 is knocked down, the fluorescence intensity of mOrange is decreased. We presented knockdown efficacy by quantifying the fluorescence intensity of mOrange relative to that of GFP. We replaced the words “CNTNAP2”, “AHI1” and “HAP1” with “CNTNAP2-mOrange”, “AHI1-mOrange” and “HAP1-mOrange”, respectively (Supplementary Fig. 2a, c and e).

(Comments)

Did you test USV's for both models or just the one?

(Our response)

We examined USV test for both CNTNAP2-knockdown and AHI1-knockdown mice (Fig. 2j and 4i).

(Comments)

P33,L4 – “open filed test” misspelling

(Our response)

We have corrected the misspelling (page 40 line 16).

(Comments)

*P33,L5 – “Mice were placed in an open field apparatus (50 x 40 x 50 cm) with 15 lx.”
What is 15 lx?*

(Our response)

“lx” stands for “lux”. We have changed lx to lux (page 37 line 4).

(Comments)

Why do reference skip from 1,2 to 9-15?

(Our response)

We have revised references from 1 to 15.

Responses to Reviewer #2:

(Comments)

This interesting manuscript addresses the hypothesis that knockdown of Cntnap2 or Ahi1 during postnatal days 16 to 24 reduces excitatory neurotransmission in the prefrontal cortex and produces autism-relevant behaviors in mice. CNTNAP2 mutations have been identified in a few people with autism spectrum disorder. AHI1 mutations are stated as causally implicated in Joubert syndrome, in which autism is associated in about 30% of cases. Studies focus on layer II/III pyramidal neurons, and test the effects of treatment with the AMPA receptor agonist CX546 on restoring electrophysiological and behavioral abnormalities in the two mouse models.

The rationale for the electrophysiological experiments is well described. Appropriate controls for the knockdowns were used, Cntnap2-scramble and Ahi1-scramble mice. The authors are to be complimented for conducting a large number of behavioral assays to investigate phenotypes relevant to the diagnostic and associated symptoms of autism.

Unfortunately, serious conceptual flaws in the methods make the key behavioral results uninterpretable.

(Our response)

We appreciate Reviewer #2's positive evaluation of our work and thank her/him for constructive suggestions and careful corrections. Our point-by-point responses to the comments are as follows.

(Comments)

Major issues

1. Social approach test

a) This test is supposed to measure time spent with a novel object versus time spent with a novel mouse. Because both wire cages were present during habituation in the present experiments, the subsequent test session compared time spent around a familiar object

versus time spent around a novel mouse. Most mice prefer novelty. As conducted, the assay measured novelty detection, not sociability.

(Our response)

We thank the reviewer for raising an important issue. As suggested by the reviewer, we examined a novel object test in both CNTNAP2-knockdown and AHI1-knockdown mice to measure novelty detection for an object. No difference was found in exploration time for the novel object (Supplementary Fig. 7 and 14). We also performed the three-chamber test to compare time spent around a novel mouse versus time spent around a novel object. We found that both CNTNAP2-knockdown and AHI1-knockdown mice did not show preference to explore the novel mouse over the novel object whereas control mice exhibited clear preference to the novel mouse (Fig. 2e-h, 4d-g, Supplementary Fig. 8b and 15b). These results suggest that both CNTNAP2-knockdown and AHI1-knockdown mice were impaired in sociability. We now present and explain these data in main figures (Fig. 2e-h, 4d-g), supplementary figures (Supplementary Fig. 7, 8b, 14 and 15b), and the Results section (page 8 line 15-19 and page 11 line 23-page 12 line 3).

(Comments)

b) “Stay time” is not an informative parameter. The tracking system must be able to determine whether the subject mouse was facing the mouse cage and facing the empty cage. The subject mouse could be in close proximity, but facing away, therefore not exploring the targets.

(Our response)

We thank the reviewer for raising an important issue. We agree with the reviewer that facing time for the mouse cage is a more appropriate parameter than stay time around the mouse cage. We measured time facing the mouse cage by an observer blind to mouse treatment conditions. Similar to the stay time, both CNTNAP2-knockdown and AHI1-knockdown mice exhibited a significant reduction in the time facing the mouse cage. The data are presented in Fig. 2d, 2g-h, 4c, 4f-g, 6e and 6k.

(Comments)

c) Statistical comparisons are incorrectly conducted. The meaningful comparison is time spent with the novel mouse versus time spent with the novel object, WITHIN genotype. ...The knockdown therefore did not produce a social deficit. There is no social abnormality signal to use in evaluating drug reversal.

(Our response)

We thank the reviewer for raising an important issue. As we state in response to the reviewer’s comment 1) a, we examined three-chamber test to compare the time sniffing the cage with the novel mouse and that with the novel object. We compared sniffing time with the novel mouse versus sniffing time with the novel object within genotype. Both CNTNAP2-knockdown and AHI1-knockdown mice did not show preference to

explore the novel mouse over the novel object whereas control mice exhibited clear preference to the novel mouse (Fig. 2e-h, 4d-g, Supplementary Fig. 8b and 15b). The results suggest that there is social abnormality signal in both CNTNAP2-knockdown and AHI1-knockdown mice.

We also performed the reciprocal social interaction test. We found that both CNTNAP2-knockdown mice and AHI1-knockdown mice exhibited a significantly smaller number of total contacts, following, nose-to-anogenital sniffing or nose-to-nose sniffing with a freely moving stranger mouse compared with CNTNAP2-scramble mice and the AHI1-scramble mice (Fig. 2i, 4h, Supplementary Fig. 8c and 15c). CX546 significantly improved the abnormal parameters seen in both CNTNAP2 knockdown and AHI1 knockdown mice (Fig. 6). Therefore, we conclude that CNTNAP2 knockdown and AHI1 knockdown produced social deficits that were rescued by administration of CX546.

(Comments)

2. Reciprocal social interactions

a) “Number of active contacts” is insufficiently explained. Freely moving mouse dyads engage in following, nose-to-anogenital sniffing, nose-to-nose sniffing, and physical contact. Each parameter should be separately scored and displayed in separate graphs.

(Our response)

We appreciate the reviewer’s helpful suggestions. We blindly scored following, nose-to-anogenital sniffing, nose-to-nose sniffing, and physical contact. CNTNAP2 knockdown mice exhibited a significant reduction in the number of following, nose-to-anogenital sniffing and nose-to-nose sniffing with a freely moving stranger mouse (Supplementary Fig. 8c). On the other hand, AHI1 knockdown mice displayed a significant decrease in the number of nose-to-nose sniffing with a freely moving stranger mouse (Supplementary Fig. 15c). In pharmacological experiments, all of the abnormal parameters seen in CNTNAP2-knockdown and AHI1-knockdown mice were restored by administration of CX546 (Fig. 6f and l). We now present and explain these data in Fig. 2i, 4h, 6f and 6l and in the Results section (page 8 line 21-22, page 12 line 4-5).

(Comments)

b) No information is provided about scoring methods by the human observers.

c) No information is provided about the specific methods used to ensure that the human observers remained blind to genotype and treatment condition while scoring.

(Our response)

We thank the reviewer for raising these important points. Each parameter in a reciprocal social interaction test was scored manually by an observer who remained blind to mouse genotype and treatment conditions. We present additional information in the Methods

section (page 37 line 8-12). A blinded observer measured the number of contacts, following, nose-to-anogenital sniffing, nose-to-nose sniffing, physical contact and contacting time. We present data in the Results section (Fig. 2i, 4h, 6f, l, Supplementary Fig. 8c, 15c, 17b and d) (page 8 line 20-23, page 12 line 3-6) and additional information in the Method section (page 37 line 8-12).

(Comments)

d) Although a significant difference was detected in number of ultrasonic vocalizations between genotypes, the magnitude of the differences was small, and unlikely to be biologically important.

(Our response)

We appreciate the reviewer's helpful suggestions. We agree that differences in USV were significant but small. In the text, we have changed the expression of "mildly impaired in social communication" or "abnormal communication" to more modest description (page 9 line 8-9, page 12 line 9-11).

(Comments)

e) Ultrasonic vocalizations emitted by separated pups are analogous to babies crying, which elicit responses by the parents and thereby meet the definition of communication. In autism, communication deficits are primarily in speech and interactive conversation during childhood and adulthood, not abnormal infant crying. Although the pup vocalization assay has been frequently used by molecular genetics labs working with mouse models of autism, it does not provide a measure of communication with high relevance to autism.

Ultrasonic vocalizations in juvenile and adult mice during social interactions offer better measures for mouse models of autism. However, the communication function of USVs in juvenile and adult mice has not been definitively established. Mice appear to communicate primarily through social odors.

The authors are encouraged to read the original literature on mouse social and communication tasks relevant to autism, and adjust their methods for future studies.

(Our response)

We appreciate the reviewer's helpful suggestions. We add these points about communication tasks including USVs in the Discussion section (page 16 line 19-page 17 line 4)

(Comments)

3. Self-grooming

As above, no information is provided about scoring methods, definitions of grooming, and methods to ensure blind rating.

(Our response)

We define grooming as (1) strokes around nose, face, ears and head with the forepaws, (2) scratches of body with the limbs or (3) body licking. Grooming time was measured by a blinded observer. We have now added this information in the Method section (page 38 line 7-9)

(Comments)

4. Light/dark box

The two parameters relevant to anxiety-related behaviors in this assay are (a) time spent in the light chamber and (2) number of transitions between the light and dark chambers. Latency to the first move into the light box, and the total distance traveled, can be removed from the data presentations.

(Our response)

We appreciate the reviewer's helpful suggestion. We have removed the latency to the first move into the light box and the total distance traveled from the presented data. Instead, we present the time spent in the light chamber and the number of transitions between the light and dark chambers (Supplementary Fig. 9e and 16e).

(Comments)

5. Forced swim

Usually the first 5 minutes in the water are excluded from scoring.

(Our response)

Followed the reviewer's helpful advice, we excluded the first 5 minutes in the water and scored immobility time for 5 minutes after the first 5 minutes. No difference was found in immobility time (Supplementary Fig. 9h and 16h). We have revised the explanation of forced swim in the Methods section (page 41 line 9-11).

(Comments)

6. Y-maze

Page 34 methods state "If a mouse consecutively enters one of the three arms with no repetitions, it was defined as alternation." This seems incorrect. Did the authors mean to say "If a mouse consecutively enters EACH of the three arms with no repetitions it was defined as alternation."?

(Our response)

We have corrected the word “one” to “each” (page 41, line 17).

(Comments)

7. AMPA drug treatment

Methods page 34 states that CX546 was administered “at a dose of 25 mg per kg or vehicle (25% b-cyclodextrin in water) 30 min before the reciprocal social interaction test or 20 min before the social novelty test.”

(a) Please justify the choice of the dose used. Describe and cite the literature on behavioral effects of CX546 in mice.

(Our response)

The dose of CX546 was based on the previous literature (Le et al., *Anesthesiology* **121**, 1080-1090, 2014, Silverman et al., *Neuropharmacology* **64**, 268-282, 2013). We added this information in the Methods section (page 41 line 23).

- Le, A.M., Lee, M., Su, C., Zou, A., and Wang, J. (2014). AMPAkinases have novel analgesic properties in rat models of persistent neuropathic and inflammatory pain. *Anesthesiology* 121, 1080-1090.
- Silverman, J.L., Oliver, C.F., Karras, M.N., Gastrell, P.T., and Crawley, J.N. (2013). AMPAKINE enhancement of social interaction in the BTBR mouse model of autism. *Neuropharmacology* 64, 268-282.

(Comments)

(b) Please justify the choice of acute treatment. Describe and cite the literature on the effects of acute versus chronic CX546 treatment on behaviors in mice.

(Our response)

Acute treatment protocol was chosen because previous studies showed that impaired social interaction was improved by acute injection of AMPA receptor agonist or positive allosteric modulator in ASD mouse models (Silverman et al., *Neuropharmacology* **64**, 268-282, 2013, Kim et al., *Neuropsychopharmacology* **44**, 314-323, 2019). In addition, acute CX546 treatment has been used for restoring pain, schizophrenic phenotype and sociality in various disease model mice (Le et al., *Anesthesiology* **121**, 1080-1090, 2014, Lipina et al., *Neuropsychopharmacology* **32**, 745-756, 2007, Silverman et al., *Neuropharmacology* **64**, 268-282, 2013). However, there were few reports for chronic treatment as far as we know. We added this information in the Methods section (page 41 line 24-page 42 line 2).

- Le AM, Lee M, Su C, Zou A & Wang J. AMPAkinases have novel analgesic properties in rat models of persistent neuropathic and inflammatory pain. *Anesthesiology* **121**, 1080-1090 (2014)

- Lipina T, Weiss K & Roder J. The ampakine CX546 restores the prepulse inhibition and latent inhibition deficits in mGluR5-deficient mice. *Neuropsychopharmacology* **32**, 745-756 (2007)
- Silverman JL, Oliver CF, Karras MN, Gastrell PT & Crawley JN. AMPAKINE enhancement of social interaction in the BTBR mouse model of autism. *Neuropharmacology* **64**, 268-282 (2013)
- Kim JW. et al. Pharmacological modulation of AMPA receptor rescues social impairments in animal models of autism. *Neuropsychopharmacology* **44**, 314-323 (2019)

(Comments)

Minor issues

1. English grammar and spelling requires corrections throughout the text. As examples:

(a) Second sentence of Abstract: “Many genes with mutation in ASD patients have been identified, ... ”

(b) Page 34 “Statistic,” “Student’s t-test was used to the behavioral data.”

(Our response)

We appreciate the reviewer’s critical reading. We carefully checked the manuscript and corrected errors.

(Comments)

2. The end of the Introduction repeats the findings stated in the Abstract and Discussion. A better use of the Introduction would be to describe the literature on CNTNAP2 mutations in people with autism, the symptoms of people with AHI1 mutations, and the existing literature on mouse models.

(a) More detailed information on the behavioral phenotypes of Cntnap2 and Ahi1 mice would be useful. There has been considerable controversy around failures to replicate the social deficits originally reported (reference 3, Peñagarikano et al.) in Cntnap2 mice, probably due to using lines from different sources and different generations of backcrosses.

(b) Literature from other mouse models of autism on excitatory and behavioral abnormalities, and on Ampakine treatments, would be useful to describe and cite in the Introduction.

(c) The human CNTNAP2 literature primarily describes polymorphisms. The original studies cited were restricted to an Amish population in Pennsylvania, USA. Later studies implicated CNTNAP2 in other disorders including speech and intellectual disabilities.

(Our response)

We thank the reviewer for helpful suggestions. We added the information about the behavioral phenotypes of CNTNAP2- and AHI1-knockdown mice, literature from other mouse models of autism on excitatory and behavioral abnormalities, and on ampakaine treatments, and literature for CNTNAP2 polymorphisms in this introduction (page 3 line 19-page 4 line2 and page 4 line 7-9).

(Comments)

Figure 1: Please remove the diagrams shown in panels b, e, g. These are standard tasks, which do not require illustrations. The diagrams are charming but unnecessary. Panel a could remain, since the authors used a non-standard apparatus to measure social approach.

(Our response)

Following the reviewer's helpful advice, we have removed the panels from Fig. 1.

Responses to Reviewer #3:

(Comments)

*This manuscript reports that reduced excitatory synaptic transmission in layer II/III pyramidal neurons in the prefrontal cortex (PFC) in two independent mouse models of ASD is responsible for the reduced social interaction. In support of these conclusions, the authors demonstrate that the I/E ratio in layer II/III pyramidal neurons in mice knocked down of *Cntnap2* and *Ahi1* by in utero electroporation is increased. This synaptic phenotype is associated with suppressed social interaction, which is rescued by CX546 (ampakine) that promotes excitatory synaptic transmission in the two independent mice.*

Given that E/I imbalance has been implicated in ASD, and mPFC in the brain, a brain region associated with higher brain functions, has not been extensively studied, these results provide significant insights into the how altered E/I balance in the mPFC could affect social behavior in mouse models of ASD.

(Our response)

We appreciate Reviewer #3's positive evaluation of our work and thank her/him for constructive suggestions. Our point-by-point responses to the comments are as follows.

(Comments)

Major comments:

1. Fig. 5 evaluates the impacts of CX546 on the decay kinetics and charge transfer of mEPSCs but does not test whether the E/I ratio, likely the most important parameter, is normalized.

(Our response)

We thank the reviewer for raising an important issue. We examined the effects of CX546 on I/E ratio in the layer II/III pyramidal neurons of the mPFC. We found no difference in I/E ratio between CX546-untreated and -treated neurons for control neurons (Fig. 5 g-h and o-p). In contrast, I/E ratio was significantly decreased in CNTNAP2 or AHI1 knockdown neurons treated by CX546 compared with those untreated by CX546 (Fig. 5 g-h and o-p), suggesting that CX546 normalizes I/E ratio. We now mention this point in the Results section (page 13 line 7-11 and 22-24) and present the data in Fig. 5g-h and 5o-p.

(Comments)

2. In addition, does CX546 have any effects on mIPSCs in layer II/III pyramidal neurons? GABAergic neurons have excitatory synapses on their dendrites, and thus it is likely that CX546 also affects the E/I balance and output functions in these neurons.

(Our response)

We thank the reviewer for this suggestion. We examined the effects of CX546 on mIPSC in layer II/III pyramidal neurons of the mPFC. There was no difference in mIPSC between CX546-untreated and -treated neurons for control, CNTNAP2-knockdown, and AHI1-knockdown neurons (Fig. 5d-f and l-n), suggesting that CX546 did not directly affect inhibitory transmission. We mention this point in page 13 and line 3-6 and present data in Fig. 5d-f and l-n.

(Comments)

Minor comments:

1. It is unclear why the authors focused on layers II/III but not other cortical layers such as layer V.

(Our response)

We appreciate the reviewer's helpful comments. Layer II/III was chosen because (1): CNTNAP2 knockout mice and other mouse models of ASD displayed synapse dysfunction and abnormal network activity in layer II/III pyramidal neurons of the cortex (Peñagarikano et al., *Cell* **147**, 235–246, 2011, Santini et al., *Nature* **493**, 411-415, 2013) and (2): ASD patients had abnormal spine morphology in layer II of the prefrontal cortex (Brodmann area 9) (Hutsler and Zhang, *Brain Res* **1309**, 83-94, 2010). In response to the similar comment from Reviewer #1, we added previous studies of layer II/III cells in ASD in the introduction section (page 4 line 14).

(Comments)

2. In Fig. 1a, it is unclear whether Cntnap2 is expressed only layer II/III or other layers.

(Our response)

Following the reviewer's suggestion, we have now examined the expression of CNTNAP2 in layer I, V and VI of the PFC and found that CNTNAP2 was expressed in all layers (Supplementary Fig. 1) (page 6 line 8-9). We also examined the expression of AHI1 and found that AHI1 was expressed in all cortical layers (Supplementary Fig. 10) (page 10 line 9-11).

(Comments)

3. It is unclear how the electroporation specifically targets layer II/III pyramidal neurons but not other cortical layers. This should be clarified.

(Our response)

We thank the reviewer for raising an important issue. Previous studies demonstrate that genes are selectively expressed in layer II/III pyramidal neurons after in utero electroporation at embryonic day 14-15 (Niwa et al., *Neuron* **65**, 480-489, 2012, Bitzenhofer et al., *Front Cell Neurosci* **11**, 239, 2017, Bitzenhofer et al., *Nat Commun* **8**, 14563, 2017). In response to the similar comment from Reviewer #1, we imaged GFP positive neurons in layer I, V and VI of the PFC after in utero electroporation at embryonic day 14. We found that there were few transfected cells in layer I, V and VI of the PFC (Supplementary Fig. 4). We now present and mention these data in Supplementary Fig. 4 and in the Results section (page 6 line 21-22).

- Niwa M1, Kamiya A, Murai R, Kubo K, Gruber AJ, Tomita K, Lu L, Tomisato S, Jaaro-Peled H, Seshadri S, Hiyama H, Huang B, Kohda K, Noda Y, O'Donnell P, Nakajima K, Sawa A, Nabeshima T. *Neuron*. 2010 Feb 25;65(4):480-9. doi: 10.1016/j.neuron.2010.01.019. Knockdown of DISC1 by in utero gene transfer disturbs postnatal dopaminergic maturation in the frontal cortex and leads to adult behavioral deficits. ¥
- Bitzenhofer SH1, Ahlbeck J1, Hanganu-Opatz IL1. *Front Cell Neurosci*. 2017 Aug 14;11:239. doi: 10.3389/fncel.2017.00239. eCollection 2017. Methodological Approach for Optogenetic Manipulation of Neonatal Neuronal Networks.
- Bitzenhofer SH1, Ahlbeck J1, Wolff A1, Wiegert JS2, Gee CE2, Oertner TG2, Hanganu-Opatz IL1. *Nat Commun*. 2017 Feb 20;8:14563. doi: 10.1038/ncomms14563. Layer-specific optogenetic activation of pyramidal neurons causes beta-gamma entrainment of neonatal networks.

(Comments)

4. Fig. 1b,c. The social novelty test referred by the authors seem to be social interaction or sociality test because this test evaluates only whether the subject mouse prefers social or empty target (not old or new social target).

(Our response)

Following the reviewer's suggestion, we have changed the terminology "social novelty test" to "sociality test".

(Comments)

5. Fig. 2e,f and Fig. 4d. Total duration of social interaction in addition to contacting number should also be shown.

(Our response)

We thank the reviewer for this suggestion. We examined the total duration of social interaction in both CNTNAP2-knockdown and AHI1-knockdown mice in a reciprocal social interaction test. Both the CNTNAP2-knockdown and the AHI1-knockdown mice exhibited a significant reduction in time of contacts with a freely moving stranger mouse (Fig. 2i and 4h) (page 8 line 20-23, page 12 line 3-6). In addition, the time of contacts was improved significantly by injection of CX546 in both CNTNAP2-knockdown mice and AHI1-knockdown mice (Fig. 6f and l).

(Comments)

6. Fig. 2b,c. Is there any reason why the authors did not use the well-known three-chamber test? In addition, the way the data in Fig. 2d could also be presented in a way by comparing the stay time of M and E because M and E are competing in the same session. The same applies to other figures that show similar results.

(Our response)

We appreciate the reviewer's helpful suggestions. It might be easier to detect a significant difference for open field than three-chamber because M and E can be easily accessible in the open field which has no a partition wall. We present data for comparing the stay time between M and E (Supplementary Fig. 8a, 15a, 17a and c). We also performed the three-chamber test and found that both CNTNAP2-knockdown mice and AHI1-knockdown mice did not prefer to explore a novel mouse over a novel object (Fig. 2e-h, 4d-g, Supplementary Fig. 8b and 15b).

(Comments)

7. Fig. 2g,h. Ideally, several early postnatal stages in addition to just P7 should be used to evaluate ultrasonic vocalizations.

(Our response)

We thank the reviewer for this suggestion. P7 was chosen because the number, duration and average peak frequency of USVs are reported to reach their peaks around P7 (Yin X et al., *PLoS One* **11**, e0160409, 2016). We describe the reason why P7 was chosen and we cite the literature in the Methods section (page 37 and line 15-16).

Yin, X et al. Maternal Deprivation Influences Pup Ultrasonic Vocalizations of C57BL/6J Mice. *PLoS One*. **11**, e0160409 (2016)

(Comments)

8. CX546, which increases excitatory synaptic transmission in WT slices, does not affect social behavior in WT mice (Figs 5 and 6). This should be explained.

(Our response)

We thank the reviewer for raising an important issue. Our data are consistent with a previous study showing that social interaction was normal in B6 control mice after injection of various ampakine compounds (Silverman et al., *Neuropharmacology* **64**, 268-282, 2013). One possibility is that control mice have maximum performance for social interaction under basal excitatory synaptic transmission. Another possibility is that I/E ratio is a critical factor for social interaction because CX546 did not change I/E ratio in control neurons. We describe the point in the Results section (page 13 line 17-24).

- Silverman JL, Oliver CF, Karras MN, Gastrell PT & Crawley JN. AMPAKINE enhancement of social interaction in the BTBR mouse model of autism. *Neuropharmacology* **64**, 268-282 (2013)

(Comments)

9. The authors need to cite previous papers that investigated the roles of PFC neurons and their synaptic transmission on social behavior in ASD models (i.e. PMID: 30659288; PMID: 28768803; PMID: 25824299 etc).

(Our response)

Following the reviewer's suggestion, we now cite these papers in the Introduction section (page 4 line 14).

- Wang ZJ *et al.* Amelioration of autism-like social deficits by targeting histone methyltransferases EHMT1/2 in Shank3-deficient mice. *Mol Psychiatry*. doi: 10.1038/s41380-019-0351-2. (2019)
- Selimbeyoglu A *et al.* Modulation of prefrontal cortex excitation/inhibition balance rescues social behavior in CNTNAP2-deficient mice. *Sci Transl Med.* **9**, pii: eaah6733. doi: 10.1126/scitranslmed.aah6733. (2017)
- Liang, J. *et al.* Conditional neuroligin-2 knockout in adult medial prefrontal cortex links chronic changes in synaptic inhibition to cognitive impairments. *Mol Psychiatry* **20**, 850-859 (2015)

(Comments)

10. The following conclusion on page 8 is too strong: "The results so far indicate that Cntnap2 knockdown in layer II/III pyramidal neurons of the developing mouse PFC reduces both excitatory and inhibitory 1 synaptic inputs to these neurons, increases I/E balance, and causes ASD-like behavior."

(Our response)

Following the reviewer's suggestion, we have changed the description in the conclusion. "The results so far indicate that CNTNAP2 knockdown in layer II/III pyramidal neurons of the developing mouse PFC results in reduced excitatory and inhibitory synaptic inputs to these neurons, increased I/E balance, and impaired sociality and communication relevant to symptoms of ASD" (page 9 line 20-23).

Reviewers' comments:

Reviewer #1 (Remarks to the Author):

The authors have adequately addressed my recommendations for improvement.

- Audrey Brumback

Reviewer #2 (Remarks to the Author):

The authors have corrected minor points but have not addressed several of major concerns in the earlier review of the original manuscript, repeated below in quotes:

1. Social approach test

a) "This test is supposed to measure time spent with a novel object versus time spent with a novel mouse. Because both wire cages were present during habituation in the present experiments, the subsequent test session compared time spent around a familiar object versus time spent around a novel mouse. Most mice prefer novelty. As conducted, the assay measured novelty detection, not sociability."

The authors have not addressed this major flaw in their methods. Re-stating what was done, in greater detail, is not helpful in addressing methodological errors.

"c) Figure 4f shows that both control and Ahi1 mice spent more time with the novel mouse than with the familiar object. The knockdown therefore did not produce a social deficit in this assay."

This negative finding needs to be clearly stated. In addition, the critical comparison in the graph is obscured by various other irrelevant statistical comparison bars.

2. Reciprocal social interactions

"c) No information is provided about the specific methods used to ensure that the human observers remained blind to genotype and treatment condition while scoring."

In the manuscript and in the Response, the authors repeatedly state that scoring was done blind, but do not explain what was done to allow the investigators to remain uninformed of genotype and treatment.

"d) Although a significant difference was detected in number of ultrasonic vocalizations between genotypes, the magnitude of the differences was small, and unlikely to be biologically important."

Two other major flaws have not been addressed:

1) The novel object recognition test was incorrectly conducted. Only one object was used, placed in the center of an open field in which the subject mouse was previously habituated. The authors are encouraged to carefully read the original methods papers for each of the behavioral assays.

2) While the results of the ultrasonic vocalization analyses have been more cautiously stated on page 12, overstatements about "impaired communication" remain in other places in the manuscript, including the Abstract.

Reviewer #3 (Remarks to the Author):

The authors have fully addressed all of my review comments. I do not have any further comments.

Responses to Reviewers' Comments

Responses to reviewers:

We thank the reviewers for critically reading our manuscript again. We have responded to the comments by reviewer #2 and revised the manuscript accordingly. We have indicated the changes in the main text, methods and figure legend with red or green so that they are easily identifiable.

Response to Reviewer #1

We appreciate Reviewer #1's positive evaluation of our work.

Responses to Reviewer #2:

We appreciate Reviewer #2's critical comments.

(Reviewer's comments)

The authors have corrected minor points but have not addressed several of major concerns in the earlier review of the original manuscript, repeated below in quotes:

1. Social approach test

a) "This test is supposed to measure time spent with a novel object versus time spent with a novel mouse. Because both wire cages were present during habituation in the present experiments, the subsequent test session compared time spent around a familiar object versus time spent around a novel mouse. Most mice prefer novelty. As conducted, the assay measured novelty detection, not sociability."

The authors have not addressed this major flaw in their methods. Re-stating what was done, in greater detail, is not helpful in addressing methodological errors.

(Our response to the comments)

Since the reviewer raised an important issue in the 1st round of review, we have done the three chamber test in which we measured time spent with a novel object versus time spent with a novel mouse (Figure 2e-h and Figure 4d-g of our revised manuscript). In this test, we found the abnormality of sociality in both CNTNAP2 knockdown and AHI1 knockdown mice compared with control mice. We think that the reviewer may have missed the result of our three chamber test or misunderstood the experimental protocol of our three chamber test (Please note the protocol in page 38, line 19 - page 39, line 11). In this test, a novel mouse was placed in one wire cage and a novel object was placed in the other wire cage in the test arena. Therefore, we measured the sociality of mouse over novelty detection, which provides a clear answer to the reviewer's point in the 1st round of review. To clearly show these points, we highlighted the sentences related to the three chamber test with green in our revised manuscript (page 8 line 17-21, page 12 line 3-6, page 38 line 19 - page 39 line 11). We consistently used the term “**novel** mouse” instead of “stranger mouse” throughout the manuscript.

(Reviewer's comments)

“c) Figure 4f shows that both control and Ahi1 mice spent more time with the novel mouse than with the familiar object. The knockdown therefore did not produce a social deficit in this assay.”

This negative finding needs to be clearly stated. In addition, the critical comparison in the graph is obscured by various other irrelevant statistical comparison bars.

(Our response to the comments)

The data in Figure 4f does not indicate a negative finding. Since the reviewer raised an important issue in the 1st round of review, we have measured the sociality index to clarify whether CNTNAP2 knockdown and AHI1 knockdown mice show any abnormality of sociality (Figure 2h and Figure 4g of our revised manuscript). We found a significant reduction of the sociality index. This index has been widely used to examine the sociality in many previous papers including those recently published in **Nature Communications** (Figure 3c in Carbonell et al., **Nat Commun**, 2019; Figure 2i in Bariselli et al., **Nat Commun**, 2018; Figure 8c, f in Peter et al., **Nat Commun**, 2016). This fact strongly indicates that our analysis is appropriate. To emphasize the results of the sociality index, we mentioned explicitly “**Additionally, the sniffing time...control**

mice.” (page 8 line 20-21) and “**However, the sniffing time...control mice.**” (page 12 line 5-6) by citing the above three papers in our revised manuscript.

References

Carbonell AU, Cho CH, Tindi JO, Counts PA, Bates JC, Erdjument-Bromage H, Cvejic S, Iaboni A, Kvint I, Rosensaft J, Banne E, Anagnostou E, Neubert TA, Scherer SW, Molholm S, Jordan BA. Haploinsufficiency in the ANKS1B gene encoding AIDA-1 leads to a neurodevelopmental syndrome. **Nat Commun** 2019 Aug 6;10(1):3529. doi: 10.1038/s41467-019-11437-w.

Bariselli S, Hörnberg H, Prévost-Solié C, Musardo S, Hatstatt-Burklé L, Scheiffele P, Bellone C. Role of VTA dopamine neurons and neuroligin 3 in sociability traits related to nonfamiliar conspecific interaction. **Nat Commun** 2018 Aug 9;9(1):3173. doi: 10.1038/s41467-018-05382-3.

Peter S, Ten Brinke MM, Stedehouder J, Reinelt CM, Wu B, Zhou H, Zhou K, Boele HJ, Kushner SA, Lee MG, Schmeisser MJ, Boeckers TM, Schonewille M, Hoebeek FE, De Zeeuw CI. Dysfunctional cerebellar Purkinje cells contribute to autism-like behaviour in Shank2-deficient mice. **Nat Commun** 2016 Sep 1;7:12627. doi: 10.1038/ncomms12627.

(Reviewer’s comments)

2. Reciprocal social interactions

“c) No information is provided about the specific methods used to ensure that the human observers remained blind to genotype and treatment condition while scoring.”

In the manuscript and in the Response, the authors repeatedly state that scoring was done blind, but do not explain what was done to allow the investigators to remain uninformed of genotype and treatment.

(Our response to the comments)

We thank the reviewer for this important comment. To clearly indicate that there were no biases of data analyses by investigators, we added the following sentences: “**Unless otherwise noted, analyses were performed automatically. To analyze behavioral data in a blinded fashion, data files were shuffled by an independent person and the experimental conditions of the test mouse (with or without knockdown, and treated by vehicle or CX546) were withheld to an observer who manually analyzed data until after the analyses were complete.**” (page 37 line 21- page 38 line 2) and “**by an observer who remained**

blind to the experimental conditions of the test mouse (with or without knockdown, and treated by vehicle or CX546) throughout analysis.” (page 38 line 14-16, page 39 line 8-9, page 39 line 22-24, page 40 line 19-21 and page 42 line 10-12).

(Reviewer’s comment)

1) The novel object recognition test was incorrectly conducted. Only one object was used, placed in the center of an open field in which the subject mouse was previously habituated. The authors are encouraged to carefully read the original methods papers for each of the behavioral assays.

(Our response to the comment)

We did not perform the “novel object recognition test” but did “the novel object test” to clarify whether CNTNAP2 knockdown mice or AHI1 knockdown mice show a change of novelty exploration (Supplementary figures 7 and 14 in the revised figures). We found that there was no significant difference in object exploration time between CNTNAP2 knockdown mice and control mice (Supplementary figures 7) and between AHI1 knockdown mice and control mice (Supplementary figures 14). These results suggest that reduction of sociality in CNTNAP2 knockdown and AHI1 knockdown mice was not due to the reduction of novelty detection. The “novel object test” has been also utilized to examine whether novelty exploration is impaired in mice in previous papers including those published in **Nature Communications** (Figure 3 in Alexander et al., **Nat Commun**, 2016; Figure 1e of Sato et al., **Nat Commun**, 2012). On the other hand, the “novel object recognition test” which the reviewer mentioned is used to assess a learning ability of mice. In contrast, the “novel object test” we performed in the present study is to assess the exploratory behavior of mice.

References

Alexander GM, Farris S, Pirone JR, Zheng C, Colgin LL, Dudek SM. Social and novel contexts modify hippocampal CA2 representations of space. **Nat Commun** 2016 Jan 25;7:10300. doi: 10.1038/ncomms10300.

Sato A, Kasai S, Kobayashi T, Takamatsu Y, Hino O, Ikeda K, Mizuguchi M. Rapamycin reverses impaired social interaction in mouse models of tuberous sclerosis complex. **Nat Commun** 2012;3:1292. doi: 10.1038/ncomms2295.

(Reviewer’s comment)

2) While the results of the ultrasonic vocalization analyses have been more cautiously stated on page 12, overstatements about “impaired communication” remain in other places in the manuscript, including the Abstract.

(Our rebuttal to the comment)

We thank the reviewer for this important comment. We follow the reviewer’s comment and rephrased “impaired communication” with “**reduced** communication” (page 2 line 10 and 15, page 4 line 17 and 20, page 10 line 2, page 11 line 19, page 15 line 17, page 18 line 12, page 19 line 18 and 22, page 20 line 1, and Fig. 4 legend).

Response to Reviewer #3

We appreciate Reviewer #3’s positive evaluation of our work.

REVIEWER COMMENTS

Reviewer #1 (Remarks to the Author):

My concerns have been adequately addressed. One minor point: for figures such as Figure 6 in which the horizontal line & asterisks span >2 bars to show significance, it is unclear which comparison(s) the asterisks correspond to. I recommend considering using horizontal lines with downward tick marks to denote which specific comparisons are being made.

Reviewer #2 (Remarks to the Author):

1. The authors continue to misunderstand the nature of their error. The fundamental problem is the HABITUATION session. The empty wire cages are present DURING THEIR FIRST SESSION, IN WHICH THE SUBJECT MOUSE IS SUPPOSED TO HABITUATE TO THE NOVELTY OF THE APPARATUS, NOT TO THE NOVELTY OF THE WIRE CAGES.

As stated in their method stated on page 38:

“In the first session of the test, two empty cylindrical wire cages (10 cm in diameter, 15 cm high) were located in both side chambers. A test mouse was placed in the center chamber and allowed to explore the two wire cages for 10 min.”

Again, to explain this issue for the third time, because both wire cages were present during habituation in the present experiments, the test session with the familiar wire cage and the novel mouse was comparing time spent with something familiar (wire cage) versus time spent with something novel (mouse). Because most mice prefer novelty, the methods used were measuring novelty detection, not sociability.

2. The authors state that other publications employed a sociability index. Again, this derived index has been used by molecular labs who are not expert in mouse behavioral assays, such as the three papers cited, but using a sociability index is incorrect. Index parameters obscure internal control measures such as general exploration of both targets, which affects derived statistics such as a sociability index.

The authors did correctly compare time spent with the novel object versus time spent with the novel mouse, within genotype and within treatment group. Interpretations of the findings should focus on this comparison only. If the number of seconds spent with the novel mouse is greater than number of seconds spent with the novel object within a genotype, or within a drug dose group, then that genotype or treatment group displays significant sociability.

As explained in this referee's previous two reviews, the correct statistical analysis in Figure 4f shows that both control and Ahi1 mice spent more time with the novel mouse than with the familiar object. The knockdown therefore did not produce a social deficit in this assay.

All other statistical comparisons in Figure 4f and similar graphs should be removed.

3. Partially acceptable statements have been added about methods used to ensure that the human raters remain blind to genotype and treatment group. However, the word “shuffle” on page 37 is unexplained. What is needed is a description of exactly what was done to code the behavioral data during analyses.

4. The authors' novel object test does not provide a comparison to a non-novel object. The method used is therefore insufficient to make an interpretation of normal interest in novelty in all genotypes. Because non-social novelty detection is being used as a control for interest in social novelty, a better control procedure is required.

5. The authors did add statements to the Discussion on page 17 that pup vocalizations are more analogous to crying than to the type of social communication deficits in language and interpreting social cues which characterize autism. They added "small but significant reduction in the mean duration of calls" on page 12.

However, "reduced communication" still appears in the Abstract. Again, (a) since pup calls are not highly relevant to the type of social communication that is impaired in people with autism, (b) because the primary parameter of number of calls was not significantly lower, and (c) given the small magnitude of the difference in the secondary parameter of duration of calls, all statements about communication must be removed from the Abstract and from the concluding paragraph of the Discussion on pages 19-20.

Reviewer #3 (Remarks to the Author):

I think the authors have fully addressed the review comments of the reviewer #2. Eunjoon Kim.

Points of Revision

Responses to reviewers:

We cordially thank the reviewers for their positive evaluation of our work and constructive suggestions and comments on our manuscript. We have tried our best to address their comments and followed their suggestions as far as possible. We feel that our manuscript has been improved significantly. We have indicated the changes in the main text, methods and figure legends with red so that they are easily identifiable.

Reviewer #1 (Remarks to the Author):

(Reviewer's comments)

My concerns have been adequately addressed. One minor point: for figures such as Figure 6 in which the horizontal line & asterisks span >2 bars to show significance, it is unclear which comparison(s) the asterisks correspond to. I recommend considering using horizontal lines with downward tick marks to denote which specific comparisons are being made.

(Our response to the comments)

We appreciate the reviewer's helpful suggestion. We have followed the reviewer's advice and now present horizontal lines with downward tick marks for graphs in which the horizontal line & asterisks span >2 bars to show significance.

Responses to Reviewer #2 (Remarks to the Author):

We deeply appreciate Reviewer #2's critical comments.

(Reviewer's comments)

1. The authors continue to misunderstand the nature of their error. The fundamental problem is the HABITUATION session. The empty wire cages are present DURING THEIR FIRST SESSION, IN WHICH THE SUBJECT MOUSE IS SUPPOSED TO HABITUATE TO THE NOVELTY OF THE APPARATUS, NOT TO THE NOVELTY OF THE WIRE CAGES.

As stated in their method stated on page 38:

“In the first session of the test, two empty cylindrical wire cages (10 cm in diameter, 15 cm high) were located in both side chambers. A test mouse was placed in the center chamber and allowed to explore the two wire cages for 10 min.”

Again, to explain this issue for the third time, because both wire cages were present during habituation in the present experiments, the test session with the familiar wire cage and the novel mouse was comparing time spent with something familiar (wire cage) versus time spent with something novel (mouse). Because most mice prefer

novelty, the methods used were measuring novelty detection, not sociability.

(Our response to the comments)

We are afraid that the reviewer simply misunderstood the experimental protocol of our three-chamber test. In the first session of the test, a test mouse was habituated to the experimental apparatus with two empty cylindrical wire cages in both side chambers. We then placed a “novel mouse” in one wire cage and a “novel object” in the other wire cage. In the second session, we measured and compared the time for which the test mouse spent around a ”novel object” (not something familiar and not present in the first habituation session) and that for which the test mouse spent around a ”novel mouse” (Figure 2e-h and Figure 4d-g of our revised manuscript). The point is that both the object and the mouse are novel to the test mouse.

To make the point clearer, we have inserted the word “newly” in the sentence that describes the method for the three chamber test: “a novel object which was similar color and similar volume to the novel mouse was newly placed in the other wire cage.”(page 39, line 3).

(Reviewer’s comments)

2. The authors state that other publications employed a sociability index. Again, this derived index has been used by molecular labs who are not expert in mouse behavioral assays, such as the three papers cited, but using a sociability index is incorrect. Index parameters obscure internal control measures such as general exploration of both targets, which affects derived statistics such as a sociability index.

The authors did correctly compare time spent with the novel object versus time spent with the novel mouse, within genotype and within treatment group. Interpretations of the findings should focus on this comparison only. If the number of seconds spent with the novel mouse is greater than number of seconds spent with the novel object within a genotype, or within a drug dose group, then that genotype or treatment group displays significant sociability.

As explained in this referee’s previous two reviews, the correct statistical analysis in Figure 4f shows that both control and Ahi1 mice spent more time with the novel mouse than with the familiar (novel) object. The knockdown therefore did not produce a social deficit in this assay.

All other statistical comparisons in Figure 4f and similar graphs should be removed.

(Our response to the comments)

We thank the reviewer for these critical comments which we must disagree with. We argue that the sociality index is useful for estimating the “degree of sociality” of animals. In Fig. 4f and 4g, we show the “reduction of sociality” but not the “absence of sociality” in AHI1 knockdown mice. Although AHI1 knockdown mice spent more time around the novel mouse than the novel object (Figure 4f), AHI1 knockdown mice showed a significant reduction of the sociality

index (Figure 4g of our revised manuscript). These results indicate that AH11 knockdown mice had sociality, but **the degree of sociality was significantly “reduced” in AH11 knockdown mice** when compared to control mice. We revised the sentences related to the sociality of AH11 knockdown mice (page 12, line 4, line 13 and line 21; page 15, line 17; page 16, line 1; page 18, line 12; page 19, line 17 and line 22; page 20, line 1; page 49, line 3; page 51, line 2).

(Reviewer’s comments)

3. Partially acceptable statements have been added about methods used to ensure that the human raters remain blind to genotype and treatment group. However, the word “shuffle” on page 37 is unexplained. What is needed is a description of exactly what was done to code the behavioral data during analyses.

(Our response to the comments)

We thank the reviewer for this comment. To clarify the word “shuffle”, we revised our manuscript as follows: “To analyze behavioral data in a blind fashion, an independent person numbered randomly all of the related data files (1 to N). N is the number of the related behavioral data files (Sociality test, Three chamber sociality test, Reciprocal social interaction test, Grooming behavior, Novel object test). The experimental conditions of the test mouse (with or without knockdown, treated with vehicle or CX546) were withheld to an observer who manually analyzed the data until the analyses were complete.” (page 37 line 20-23).

(Reviewer’s comments)

4. The authors’ novel object test does not provide a comparison to a non-novel object. The method used is therefore insufficient to make an interpretation of normal interest in novelty in all genotypes. Because non-social novelty detection is being used as a control for interest in social novelty, a better control procedure is required.

(Our response to the comments)

The purpose of our novel object test was not to compare the interest to a novel object and that to a non-novel object in individual mice. We found that both CNTNAP2-knockdown mice and AH11-knockdown mice displayed a significant decrease in sniffing and stay time around a novel mouse in the sociality test in which the novel mouse was newly placed after habituation of the subject mouse to an open field. The decrease in sniffing and stay time around a novel mouse might be due to the decrease in exploratory behavior toward a newly placed object, but not due to impaired sociality. To test this possibility, **we did the novel object test in which we reproduced a situation similar to the sociality test** (an inanimate object was newly placed after habituation of the subject mouse to an open field). In this test, there was no difference between control mice and knockdown mice in the exploratory behavior toward the newly placed object. Although this test did not compare exploratory behavior between a non-novel object and a novel object, which was not the purpose of the present test, **the results indicate that both of the knockdown mice have normal**

exploratory behavior toward a newly placed object. These results strongly suggest that **the decrease in sniffing and stay time around a novel mouse in the sociality test is due to reduced sociality, but not due to decreased exploratory behavior toward a novel object.**

(Reviewer's comments)

5. The authors did add statements to the Discussion on page 17 that pup vocalizations are more analogous to crying than to the type of social communication deficits in language and interpreting social cues which characterize autism. They added “small but significant reduction in the mean duration of calls” on page 12.

However, “reduced communication” still appears in the Abstract. Again, (a) since pup calls are not highly relevant to the type of social communication that is impaired in people with autism, (b) because the primary parameter of number of calls was not significantly lower, and (c) given the small magnitude of the difference in the secondary parameter of duration of calls, all statements about communication must be removed from the Abstract and from the concluding paragraph of the Discussion on pages 19-20.

(Our response to the comments)

We carefully checked our manuscript and deleted the phrases “reduced communication”. We use **“mild vocalization abnormality”** instead (page 2, line 10 and line 15; page. 4, line 16-17 and line 19-20; page. 9, line 6 and line 11; page 9 line 24 to page 10, line 1; page 11, line 18 and line 21; page. 12, line 13-14; page. 15, line 17-18; page. 18, line 12; page. 19, line 18 and line 22; page 20, line 1; page 49, line 4).

Reviewer #3 (Remarks to the Author):

We appreciate Reviewer #3's positive evaluation of our work.

REVIEWERS' COMMENTS:

Reviewer #1 (Remarks to the Author):

The authors have addressed all of my suggestions.

Points of Revision

Responses to reviewers:

We cordially thank the reviewers for their positive evaluation of our work and many constructive suggestions and comments on our manuscript.

Reviewer #1 (Remarks to the Author):

The authors have addressed all of my suggestions.

We deeply appreciate reviewer #1's positive evaluation of our work.